# Structures of p53/BCL-2 complex suggest a mechanism for p53 to antagonize BCL-2 activity

Hudie Wei [1,5], Haolan Wang[1,5], Genxin Wang[2,3], Lingzhi Qu[1], Longying Jiang [1,4], Shuyan Dai[1], Xiaojuan Chen[1], Ye Zhang[1], Zhuchu Chen[1], Youjun Li [2,3], Ming Guo [1] ✉ & Yongheng Chen [1] ✉

Mitochondrial apoptosis is strictly controlled by BCL-2 family proteins through a subtle network of protein interactions. The tumor suppressor protein p53 triggers transcription-independent apoptosis through direct interactions with BCL-2 family proteins, but the molecular mechanism is not well understood. In this study, we present three crystal structures of p53-DBD in complex with the anti-apoptotic protein BCL-2 at resolutions of 2.3–2.7 Å. The structures show that two loops of p53-DBD penetrate directly into the BH3-binding pocket of BCL-2. Structure-based mutations at the interface impair the p53/BCL-2 interaction. Specifically, the binding sites for p53 and the pro-apoptotic protein Bax in the BCL-2 pocket are mostly identical. In addition, formation of the p53/BCL-2 complex is negatively correlated with the formation of BCL-2 complexes with pro-apoptotic BCL-2 family members. Defects in the p53/BCL-2 interaction attenuate p53-mediated cell apoptosis. Overall, our study provides a structural basis for the interaction between p53 and BCL-2, and suggests a molecular mechanism by which p53 regulates transcription-independent apoptosis by antagonizing the interaction of BCL-2 with pro-apoptotic BCL-2 family members.

Dysregulation of apoptosis has been widely described in tumorigenesis and drug resistance[1]. Activating apoptosis in cancer cells is a mainstay and goal of clinical oncology[2]. BCL-2 family proteins and the tumor suppressor protein p53 are central regulators in apoptosis signaling pathways[3,4]. p53 regulates the activity of the BCL-2 family through transcription-dependent and transcription-independent pathways[5,6]. As a nuclear transcription factor, p53 transcriptionally activates the expression of pro-apoptotic genes such as NOXA, Bax, and Puma[7]. In the cytoplasm and mitochondrial membrane, p53 also triggers apoptosis through the formation of inhibitory complexes with

the anti-apoptotic proteins BCL-2 and BCL-xL[8], or by activating the pro-apoptotic effector proteins Bak and Bax[9,10].

Pro-apoptotic and anti-apoptotic BCL-2 family proteins strictly control the mitochondrial apoptosis pathway (also called the intrinsic apoptosis pathway) through protein–protein interaction networks[11]. Bak and Bax directly control the integrity of the outer mitochondrial membrane (OMM) and the initiation of apoptosis[12]. In response to apoptotic stimuli, Bak and Bax are directly or indirectly activated by pro-apoptotic BH3-only activators or sensitizers, and form pores across the OMM, which leads to the release of cytochrome C into the

[1]Department of Oncology, NHC Key Laboratory of Cancer Proteomics & State Local Joint Engineering Laboratroy for Anticancer Drugs, National Clinical Research Center for Geriatric Disorders, Xiangya Hospital, Central South University, Changsha, Hunan 410008, China. [2]Hubei Key Laboratory of Cell Homeostasis, College of Life Sciences, TaiKang Center for Life and Medical Sciences, Wuhan University, Wuhan 430072, China. [3]Frontier Science Center for Immunology and Metabolism, Medical Research Institute, Wuhan University, Wuhan 430071, China. [4]Department of Pathology, Xiangya Hospital, Central South University, Changsha, Hunan, China. [5]These authors contributed equally: Hudie Wei, Haolan Wang. ✉e-mail: guomingxyyy@163.com; yonghenc@163.com

cytoplasm and the activation of the caspase cascade[13]. However, anti-apoptotic BCL-2 proteins antagonize the activity of pro-apoptotic proteins by canonically binding to and sequestering the BH3 domain of pro-apoptotic proteins through a large hydrophobic pocket (called the BH3-binding pocket)[14]. This canonical interaction mechanism promotes the development of BH3-mimetic drugs, which selectively inhibit anti-apoptotic BCL-2 members by competitively occupying the BH3-binding pocket[15].

p53 plays a crucial role in the prevention of cancer development[16]. In approximately 50% of human cancers, p53 function is abrogated by gene mutations[17]. The direct interactions of p53 with the anti-apoptotic proteins BCL-2 and BCL-xL have important roles in the p53-mediated mitochondrial apoptosis pathway[5,8,18]. The activities of p53 depend on its DNA-binding domain (p53-DBD)[8]. p53-DBD binds BCL-xL at a surface distinct from the BH3 binding pocket, as demonstrated by NMR and crystal structure studies[19–21]. Our previous crystal structure of the p53-DBD/BCL-xL complex showed that p53-DBD forms a homodimer and binds to BCL-xL at a surface consisting of the C-terminal residues of α1 and α5, the α5-α6 loop and the α3-α4 loop[21]. BCL-2 and BCL-xL share 45% sequence identity, whereas they possess different sensitivities to interactions with pro-apoptotic proteins[22].

In this study, we determine three crystal structures of the p53-DBD/BCL-2 complex. To our surprise, p53-DBD interacts with BCL-2 in a very different manner than that observed for BCL-xL in our previous structure[21]. Based on these structures, mutational and functional analyses elucidate the molecular mechanism of the interaction between p53 and BCL-2, and reveal the regulatory mechanism of the p53/BCL-2 interaction on pro-apoptotic BCL-2 family proteins.

## Results

### Crystal structures of the p53-DBD/BCL-2 complex

Human p53 is a multi-domain protein that consists of an N-terminal transactivation domain (NTD, residues 1-92), a core DNA-binding domain (DBD, residues 96-292) linked to a tetramerization domain (TET, residues 324-356), and a C-terminal domain (CTD, residues 364-393)[23] (Fig. 1a). The DBD is the main domain responsible for binding BCL-2[8,24]. Human BCL-2 contains four BCL-2 homologous domains (BH1-4) and a C-terminal transmembrane domain (Fig. 1a, b).

To investigate the detailed interactions between p53 and BCL-2, we performed crystallographic structural analysis. An initial trial to crystallize the p53-DBD/BCL-2 complex by mixing p53-DBD with BCL-2 proteins failed. To stabilize the p53-DBD/BCL-2 complex, we used a fusion strategy (Fig. 1c). First, BCL-2 was linked to p53-DBD via a glycine-rich linker that was previously used for the p53-DBD/BCL-xL complex to increase the proximity between the two proteins[21]. However, we noted that a BCL-2 construct (BCL-2#3), featuring a shortened α1-α2 loop and certain surface residues replaced with a BCL-xL sequence (Fig. 1a, b), could be expressed in high yield and was suitable for crystallization[25]. BCL-2#3 was reported to have a similar selectivity profile for BH3-only peptides as wild-type BCL-2 and a weak interaction with the BCL-xL-specific peptide Hrk[25]. Truncation strategies have been used in many biochemical and structural studies of the BCL-2 family (Fig. 1b), such as BCL-2#2 for solution NMR of BCL-2[26], crystal structures of BCL-2/Bax-BH3[27] and BCL-2/venetoclax[28], BCL-2#3 for crystal structure of BCL-2/Puma-BH3[25] and so on. To improve the success of crystallization and to ensure that the linker does not affect protein interactions, we linked this BCL-2#3 construct to p53-DBD via three different glycine-rich linkers (Fig. 1c)[29]. Gel filtration chromatography analysis revealed that the fusion proteins were highly homogeneous (Supplementary Fig. 1). In addition, the microscale thermophoresis (MST) assay showed that p53-DBD had similar KD values for both BCL-2 (3.5 μM) and BCL-2#3 (3.9 μM), suggesting that the modifications of BCL-2 did not influence its binding affinity to p53-DBD (Supplementary Fig. 2).

Among these crystallization attempts, small and needle-like crystals can be obtained for BCL-2 fusion proteins, but it is difficult to optimize them for X-ray diffraction. In particular, crystals of the three BCL-2#3-linker-p53-DBD fusion proteins were successfully optimized for X-ray diffraction, and determined at resolutions of 2.3–2.7 Å in the P $2_1 2_1 2_1$ space group (Supplementary Table 1). The RMSDs for superposition of these three structures are 0.2–0.5 Å (Fig. 1d). p53-DBD and BCL-2#3 had similar interaction modes in these three structures, indicating that the linkers did not affect the complex structure. We henceforth used the 22 residue-linked complex structure as a representative structure for the detailed analysis (Fig. 1e). In the p53-DBD/BCL-2#3 complex structure, the p53-DBD molecule shares a similar overall structure to the apo structure of p53-DBD (PDB: 2OCJ)[30] with a RMSD of 0.4 Å (Supplementary Fig. 3a). Two loops of p53, the loop linking strands S5 and S6 (S5-S6 loop) and the large loop L2, insert into the BH3-binding pocket of BCL-2#3 which is composed of BH1-3 domains (Fig. 1e–g). The overall structure of BCL-2#3 closely resembles the apo solution structure of BCL-2 (PDB: 1GJH)[26], with small conformational changes in the α3 and α4 regions and in the C-terminus of α2 and α5 (Supplementary Fig. 3b). Notably, the flexible loop and modified residues of BCL-2 are not located on the contact surface with p53-DBD (Supplementary Fig. 3c). This indicates that the modifications of BCL-2 have minimal influence on the complex structure.

### Detailed interactions in the p53-DBD/BCL-2 complex structure

The p53-DBD/BCL-2#3 interface covers approximately 1050 Å$^2$ and encompasses a hydrophobic core and surrounding polar interactions (Fig. 2, Supplementary Fig. 4). The involved BCL-2 residues are mainly from the α2-α3 and α4-α5 corner regions. Two hydrophobic residues of p53, Leu188 and Leu201, penetrate deeply into the hydrophobic pocket of BCL-2 and interact with BCL-2 residues Phe104, Tyr108, Phe112, Met115 and Leu137 (Fig. 2a). Around the hydrophobic core, Arg107 and Tyr108 of BCL-2 form water-mediated hydrogen bond or direct hydrogen bond contacts with p53 residue Asp186 (Fig. 2b, Supplementary Fig. 4). Asp111, Asn143 and Arg146 of BCL-2 form direct hydrogen bonds with the backbone atoms of residues Leu201, Gly187 and Val203 of p53, respectively (Fig. 2b). In addition, several van der Waals force interactions help stabilize the p53-DBD/BCL-2#3 interactions (Fig. 2c).

### Mutational analysis of the p53/BCL-2 interactions in vitro

To verify the p53-DBD/BCL-2 interface, we performed structure-based mutational analysis. The key interfacial residues of BCL-2, including Phe104, Arg107, Tyr108 and Arg146, and the key interfacial residues of p53, including Asp186, Leu188 and Leu201, were mutated and tested (Fig. 3).

We first used pull-down assays to detect the effect of these mutations on p53-DBD/BCL-2 interactions. These BCL-2 mutations were introduced into recombinant BCL-2 constructs. His-tagged BCL-2 or its mutants were incubated with p53-DBD proteins, and then nickel beads were used to capture the complex (Fig. 3a). Compared to wild-type BCL-2, the complex formation of p53-DBD with BCL-2 mutants F104A, R107A, Y107A and R146A was reduced. Accordingly, GST-tagged BCL-2 was incubated with p53-DBD or its mutants and then captured by glutathione sepharose beads. The p53-DBD mutants D186R, L188R and L201R had a weaker ability to bind to GST-tagged BCL-2 (Fig. 3b). ELISA analysis was also carried out to detect the effects of these mutations. His-tagged BCL-2 proteins were captured by a 6 × His-tag antibody-conjugated well plate and then incubated with a gradient dilution of GST-p53-DBD protein. After multiple washes, the bound p53 was quantified by the quantity of a GST detector antibody. Compared to the wild-type, the BCL-2 mutants were defective in the ability to capture p53-DBD protein (Supplementary Fig. 5). Additionally, p53-DBD mutants were defective in the interaction with BCL-2. By

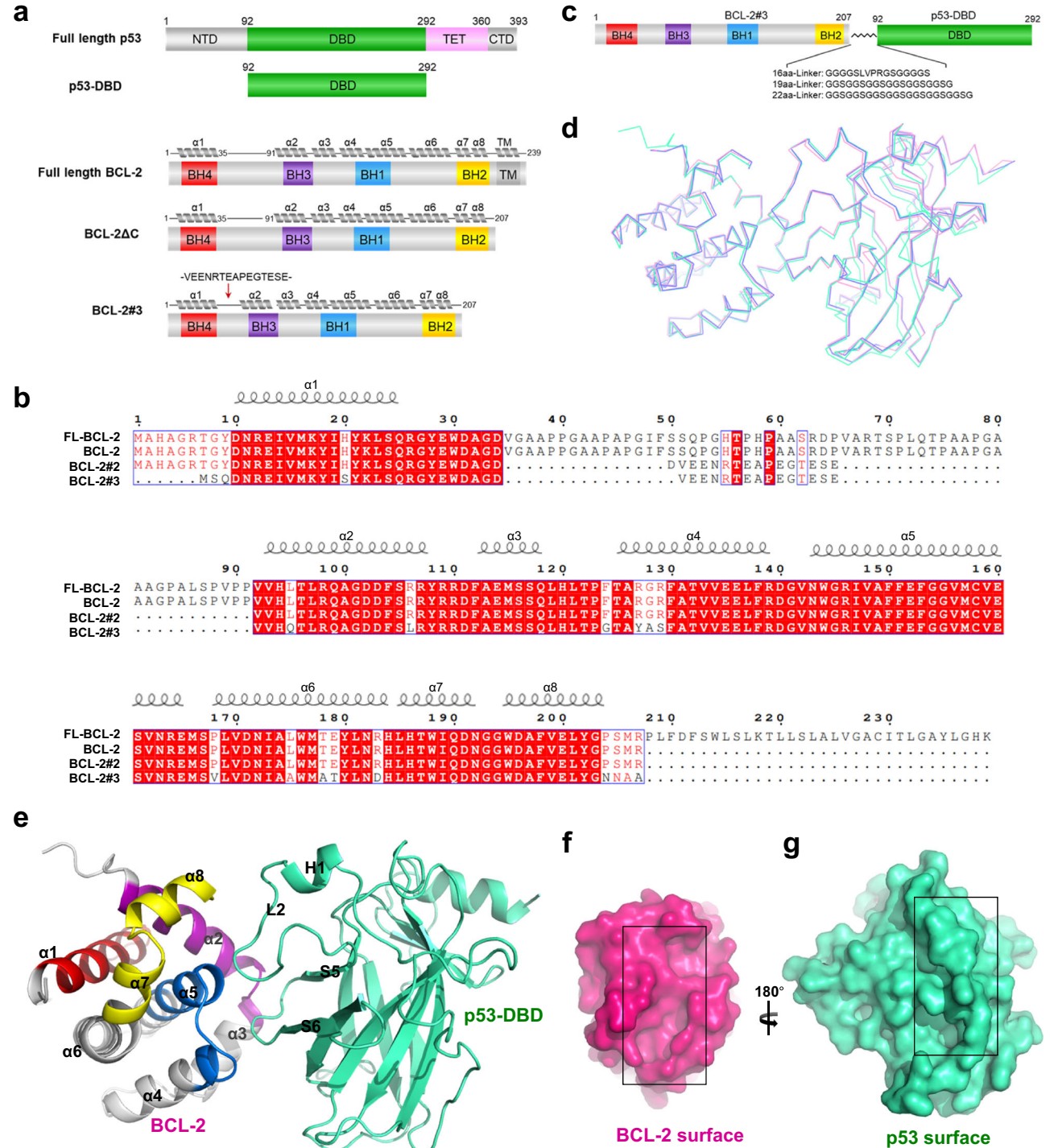

**Fig. 1 | Crystal structures of the p53-DBD/BCL-2 complex. a** Schematic representation of p53 and BCL-2 constructs. The p53 and BCL-2 constructs are shown as indicated. **b** Sequences of BCL-2 constructs. The sequence alignment was created using the web services ESPript 3.0. Identical and similar residues are boxed in red and white, respectively. **c** Schematic representation of the linked BCL-2-p53-DBD constructs. The sequences of glycine-rich linkers are listed as indicated. **d** Superposition of the three structures of the BCL-2/p53-DBD complex. Structures of 16, 19 and 22 residue-linked complexes are shown as pink, slate and green ribbons, respectively. **e** Cartoon presentation of the p53-DBD/BCL-2 structure. BCL-2 is shown as gray cartoons with BH1-BH4 colored blue, yellow, purple and red, respectively. p53 is colored green. **f** Surface presentation of the BCL-2 molecule. **g** Surface presentation of the p53 molecule by rotating 180° along the y-axis of the view in (**f**). The black box indicates the interface.

analyzing the experimental data with a sigmoidal dose-response non-linear regression model, it was shown that all these mutants had a higher EC50 (Supplementary Fig. 5).

MST assays were performed to quantify the binding affinity (Fig. 3c, d). A gradient dilution of GST-p53-DBD proteins was titrated against RED-Tris-NTA dye-labelled His-BCL-2. The KD values of wild-type p53-DBD binding to BCL-2 mutants F104A, R107A, Y108A and R146A were 42, 28, 26 and 21 μM, respectively (Fig. 3c). The KD values of p53-DBD mutants D186R, L188R and L201R binding to wild-type BCL-2 were 8.9, 14 and 7.6 μM, respectively (Fig. 3d). A p53-DBD

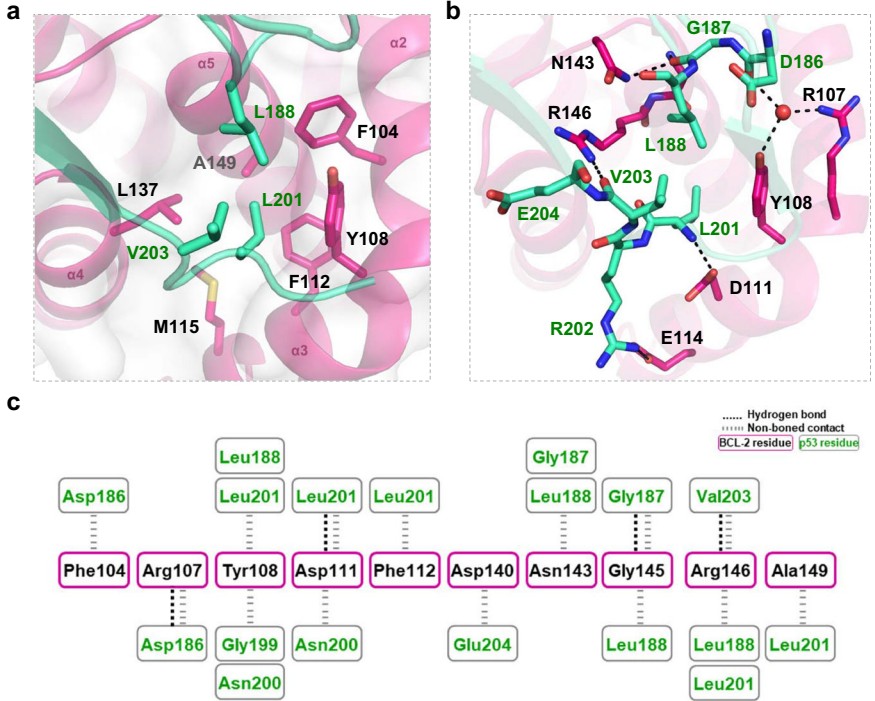

**Fig. 2 | Detailed interactions at the p53-DBD/BCL-2 interface. a** Hydrophobic core at the p53-DBD/BCL-2 interface. BCL-2 is shown as hot pink cartoons and white surface. p53 is shown as green cartoons. The key residues are shown as sticks. BCL-2 residues and p53 residues are labelled black and green, respectively. **b** Polar contacts around the hydrophobic core. Hydrogen bonds and salt bridges are shown as dark dashed lines. Water is shown as red spheres. **c** Schematic diagram of the detailed contacts at the p53-DBD/BCL-2 interface.

mutant carrying three residue mutations (D186R/L188R/L201R, called RRR) showed a KD of 11 μM. Compared with the wild-type (KD = 3.5 μM), the BCL-2 mutants or p53-DBD mutants had a decreased affinity. Overall, the pull-down and binding affinity data demonstrated that these structure-guided mutants of BCL-2 or p53-DBD disrupted the p53-DBD/BCL-2 interactions, and supported the interface defined by the p53-DBD/BCL-2#3 complex structure.

### p53 and pro-apoptotic Bax bind at the same pocket of BCL-2

A typical interaction mechanism of the BCL-2 family is that the BH3-binding pocket of anti-apoptotic proteins occupies the BH3 domain of pro-apoptotic proteins, thus blocking the pro-apoptotic activity[31]. Pro-apoptotic Bax is an effector protein that directly controls mitochondrial outer membrane permeabilization (MOMP) and initiation of apoptosis[32]. BCL-2 inhibits Bax activation via its tight and preferential interaction with the BH3 domain of Bax[27]. Superposition of the BCL-2 structure of the p53-DBD/BCL-2#3 complex with that of the BCL-2/Bax-BH3 complex (PDB: 2XA0)[27] shows that p53-DBD and the BH3 domain of Bax bind at the same pocket of BCL-2 (Fig. 4a). This could lead to competition and prevent either protein from binding to BCL-2.

p53 binding results in a smaller pocket, whereas Bax-BH3 binding leads to an opening pocket and less helical α3 (Fig. 4b). The Bax-BH3 peptide folds into an amphipathic α-helix with the four conserved hydrophobic residues accommodated within small hydrophobic pockets (P1-P4 pockets) of BCL-2, while some salt bridge interactions are formed on the sides (Fig. 4c). In comparison, p53 utilizes two flexible loops to form core hydrophobic interactions with BCL-2 at the corresponding P2-P3 pockets, while polar interactions are formed around the core (Fig. 4d). The hydrophobic BCL-2 residues (Phe104, Tyr108, Phe112, Met115, Leu137 and Ala149), as well as the polar residues (Arg107, Asp111, Asp140, Asn143, Gly145, Glu136 and Arg146), are involved in the interactions with both p53 and Bax-BH3 (Fig. 4e, f). In addition, the BCL-2 mutants F104A, R107A, Y108A and R146A, which

showed decreased affinity for p53-DBD (Fig. 3c), also exhibited significantly lower affinity for the Bax-BH3 peptide in the fluorescence polarization assay (FPA) (Fig. 5a). These results demonstrated the dual roles of these BCL-2 residues in binding both p53 and Bax.

### p53 inhibits BCL-2 binding to pro-apoptotic BCL-2 family proteins in vitro

To examine the possible effects of p53 on the Bax/BCL-2 interactions, we used p53-DBD to compete for BCL-2 from fluorescent Bax-BH3 in a competition FPA. p53-DBD at a concentration of 30 μM caused an approximately 60% inhibition of the BCL-2/Bax-BH3 interaction (Fig. 5b). The p53-DBD mutant L188R at 30 μM had a similar 60% inhibition, whereas the mutants D186R and L201R at 30 μM only caused 40 and 50% inhibition, respectively (Fig. 5b). The IC50 value was 738 nM for wild-type p53-DBD on the basis of a sigmoidal dose-response nonlinear regression model. The IC50 values for mutants D186R, L188R and L201R were 1.2, 6.4 and 2.1 μM, respectively. Thus, p53-DBD could liberate Bax-BH3 from the BCL-2 complex, while the p53-DBD mutants with a weaker affinity for BCL-2 were defective in this ability.

Given that several pro-apoptotic BCL-2 members are able to interact with BCL-2 through the BH3-binding pocket[12], GST pull-down experiments were performed to detect whether p53 was able to compete with other pro-apoptotic BCL-2 family members. GST-tagged p53-DBD proteins were incubated with His-tagged BCL-2 in the absence or presence of different BH3 peptides (Fig. 5c, Supplementary Table 2). The complex formation of BCL-2/p53-DBD was decreased in the presence of Bax-BH3, Bak-BH3, Bid-BH3, Bim-BH3 and Puma-BH3 peptides (Fig. 5c). Furthermore, when GST-BCL-2 proteins were simultaneously incubated with p53-DBD and recombinant Bax proteins, a decrease in BCL-2/p53-DBD complex formation correlated with an increase in BCL-2/Bax complex formation (Fig. 5d). Recombinant Bak proteins were also tested. GST-BCL-2 could pull down the Bak protein (Fig. 5e, lane 1), although BCL-2 has been reported to have a weaker interaction with Bak[33]. In the

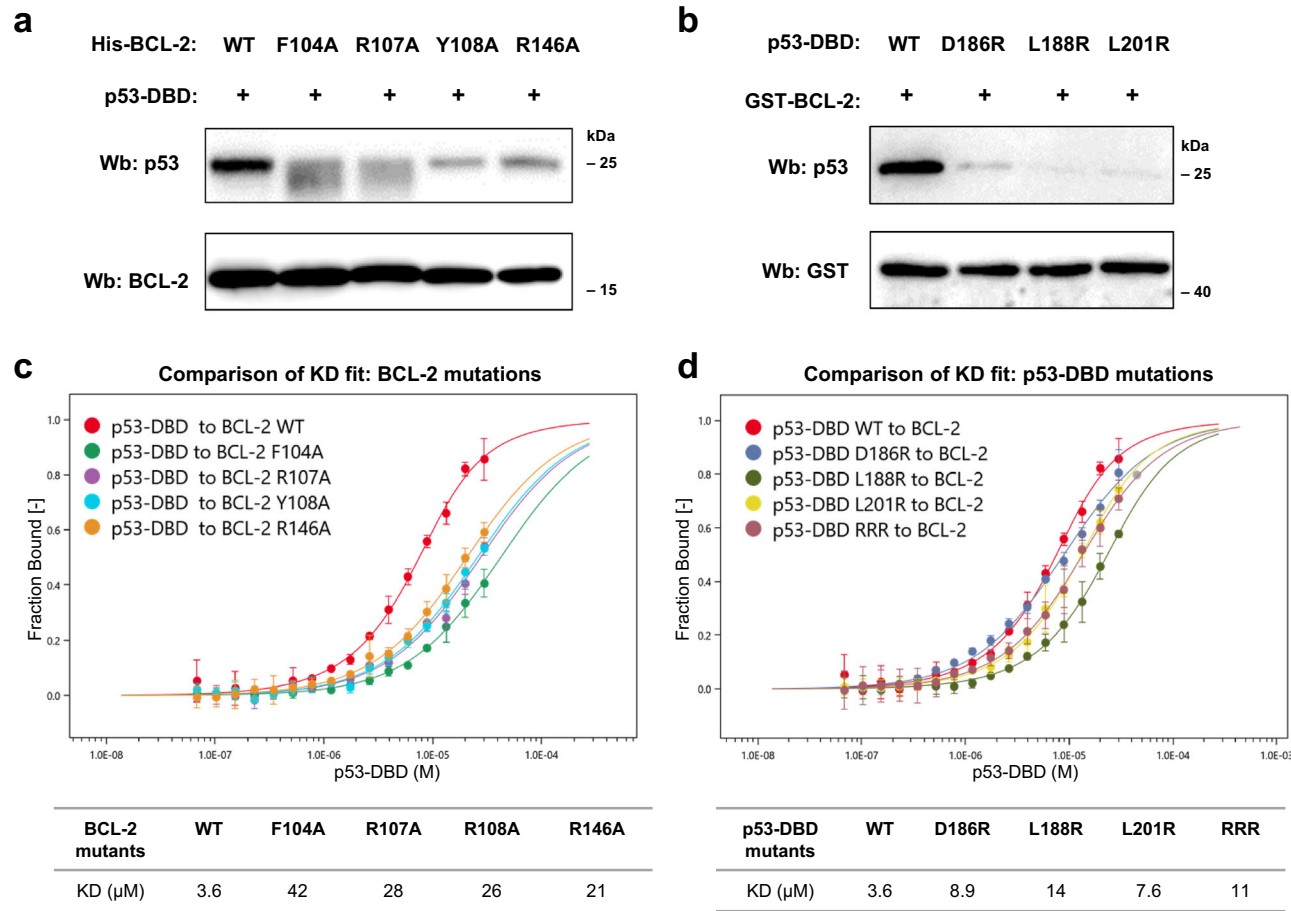

**Fig. 3 | Mutational analysis of the p53-DBD/BCL-2 interactions. a** Wild-type or mutant His-BCL-2 proteins were incubated with wild-type p53-DBD. The complex was immobilized by nickel beads and subjected to western blotting as indicated. **b** Wild-type GST-BCL-2 was incubated with wild-type or mutant p53-DBD. The complex was immobilized by glutathione sepharose and subjected to western blotting as indicated. **c** The binding affinity of wild-type GST-p53-DBD to BCL-2 mutants was analyzed by MST. **d** The binding affinity of GST-p53-DBD mutants to wild-type BCL-2 was analyzed by MST. Data are represented as the mean ± SD of $n = 3$ independent experiments. Curves were fitted, and KD values of interactions were calculated using experimental data with a KD model by MO. Affinity Assay software. Source data are provided as a Source Data file.

presence of p53-DBD, the BCL-2/Bak interaction was also inhibited (Fig. 5e).

Overall, these results indicated that p53-DBD had a competitive correlation with these pro-apoptotic BCL-2 family members. However, p53-DBD cannot fully displace Bax or Bak proteins from GST-BCL-2 even when the amount of p53-DBD is approximately 10 times higher than that of Bax (Fig. 5d, lane 5) or Bak proteins (Fig. 5e, lane 5). This is consistent with the competition FPA data that p53-DBD cannot lead to complete inhibition of BCL-2/Bax-BH3 (Fig. 5b). It is possible that other interactions may hold the complex in place.

**Mutations at the p53/BCL-2 interface inhibit p53-mediated cell apoptosis**

The results of structural and mutational analysis in vitro elaborated the interaction mode of p53-DBD binding to recombinant BCL-2. To verify the cellular interaction of full-length p53 (FL-p53) with full-length BCL-2 (FL-BCL-2), coimmunoprecipitation assays were carried out in HEK 293 T cells. Wild-type flag-tagged FL-p53 showed predominant complex formation with HA-tagged BCL-2, whereas the mutant FL-p53[RRR] (D186R/L188R/L201R), in which the three key residues at the p53/BCL-2 interface were mutated, was defective in coimmunoprecipitation with FL-BCL-2 (Fig. 6a). The results demonstrated the critical roles of the three residues in the cellular interaction between full-length p53 and BCL-2, in agreement with the structural analysis.

The effect of mutations at the p53/BCL-2 interface on cell apoptosis was then assessed. To exclude the transcriptional apoptotic signaling of p53, we used a transcriptionally inactive form of p53, S269E/T284E[34], termed Ti-p53. The two residues Ser269 and Thr284 are away from the p53/BCL-2 interface in the complex structure. When stably expressed in p53[-/-] HCT116 cells, Ti-p53 maintained the ability to coprecipitate with FL-BCL-2, but Ti-p53[RRR] did not (Supplementary Fig. 6). In addition, Ti-p53 expression can induce the cleavage of caspase 3 and PARP, whereas p53 mutations at the p53/BCL-2 interface resulted in reduced cleavage of caspase 3 and PARP as shown by Ti-p53[RRR] compared with Ti-p53 (Fig. 6b). Flow cytometry assays and detection of an Annexin-V/7-AAD marker showed that Ti-p53 induced approximately 19.6% of cells to undergo apoptosis, whereas the ratio of apoptotic cells was reduced significantly to 8% with Ti-p53[RRR] (Fig. 6c, d, Supplementary Fig. 7). Taken together, these results indicated that p53 interacted with BCL-2 and promoted cell apoptosis, while defects in the p53/BCL-2 interaction reduced p53-mediated apoptosis.

To examine whether Bax or Bak is activated by Ti-p53 activity, we knocked down Bax or Bak by small interfering RNA (siRNA) in Ti-p53 stably expressing HCT116 cells, and then cell apoptosis was detected by flow cytometry analysis (Supplementary Fig. 8). Compared to Ti-p53 (19.6%), Bax siRNA and Bak siRNA decreased the apoptotic cells to 12.1% and 13.5%, respectively. The simultaneous knockdown of Bax and Bak still led to 9.6% of cell apoptosis. The

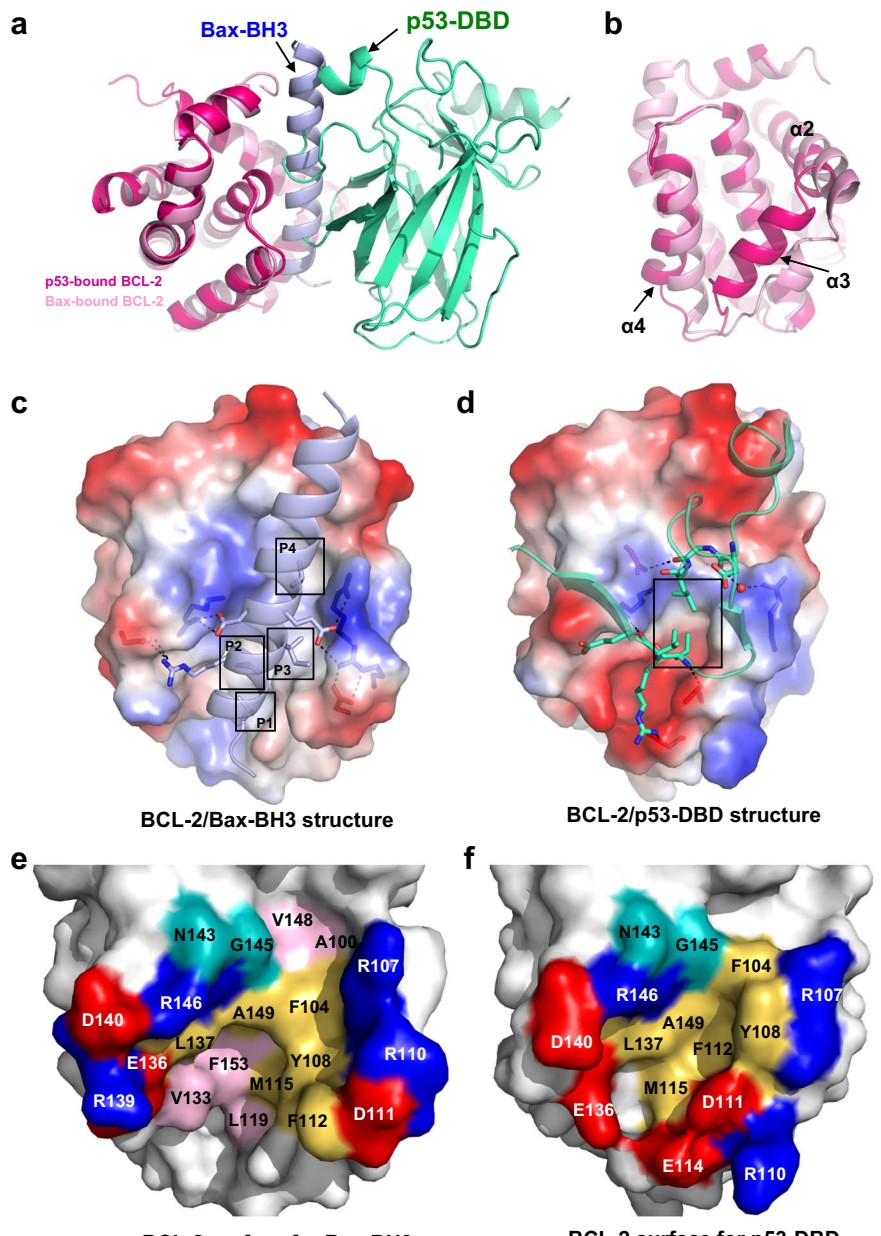

**Fig. 4 | Structural analysis shows p53 and pro-apoptotic Bax binding at the same pocket of BCL-2. a** Superposition of the BCL-2 structure in the p53-DBD/BCL-2 complex with the BCL-2 structure in the BCL-2/Bax-BH3 complex (PDB: 2XA0). BCL-2 and p53 in the p53-DBD/BCL-2 complex are colored hot pink and green, respectively. BCL-2 and Bax-BH3 in the BCL-2/Bax-BH3 complex are colored pink and light blue, respectively. **b** Superposition of BCL-2 structures shows the BH3-binding pockets. **c, d** The electrostatic surface potential of the BCL-2 molecule shows BCL-2/Bax-BH3 interactions (**c**) or BCL-2/p53-DBD interactions (**d**). The black box indicates the hydrophobic pockets of BCL-2. **e, f** The interfacial BCL-2 residues in the BCL-2/Bax-BH3 structure (**e**) or BCL-2/p53-DBD structure (**f**). Residues are colored and labelled as indicated.

results indicate that apoptosis induced by Ti-p53 is dependent on both Bax and Bak. A previous study[35] also reported that the DNA damage-induced p53/BCL-2 interaction resulted in decreased BCL-2/Bax heterodimers in H7 cells, and depletion of Bax by RNA interference blocked p53-induced cytochrome c release from isolated mitochondria of H1299 p53-null cells, which is consistent with our results.

## Discussion

The ability to induce apoptosis is among the most important biological functions of p53 in the prevention of cancer development[36]. Increasing studies have provided evidence that p53 promotes mitochondrial apoptosis through direct interactions with BCL-2 family members. In

this study, the structures of p53-DBD/BCL-2 provide insight into the interaction between p53 and the anti-apoptotic member BCL-2. Interestingly, p53 uses two flexible loops (S5-S6 loop and loop L2) to directly occupy the BH3-binding pocket of BCL-2. The binding sites of p53 in BCL-2 largely overlap with those of pro-apoptotic members of the BCL-2 family, and thus, p53 can competitively inhibit the interaction of BCL-2 with pro-apoptotic family members. Therefore, our study supports the activity of p53 as a BH3-lacking sensitizer to antagonize the anti-apoptotic activity of BCL-2 and consequently promote the release of pro-apoptotic BCL-2 family proteins and activation of apoptosis.

BCL-2 and BCL-xL are highly homologous anti-apoptotic members, and both have direct interactions with p53. Previous studies have

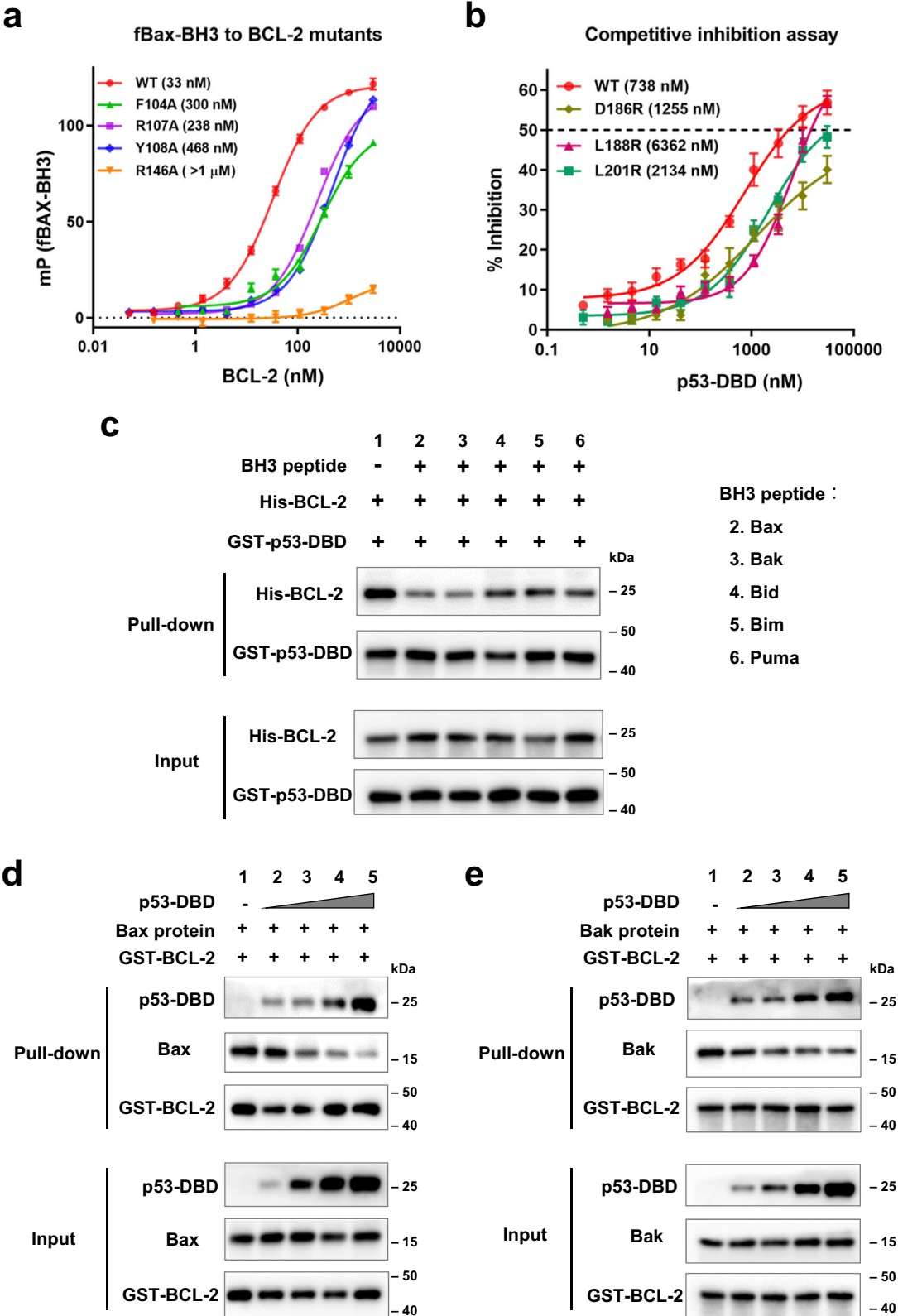

**Fig. 5 | p53 competitively inhibits BCL-2 binding to pro-apoptotic BCL-2 family proteins in vitro. a** Fluorescence polarization assay shows BCL-2 binding to fluorescein-labelled Bax-BH3 peptide (fBax-BH3). **b** Competitive inhibition assay of the p53-DBD to BCL-2/fBax-BH3 interaction. p53-DBD or its mutant was used as a competitor in the BCL-2/fBax-BH3 fluorescence polarization assay. The EC50 and IC50 values of (**a**) and (**b**) were determined by fitting the data using a sigmoidal dose-response nonlinear regression model and shown as indicated. Data are represented as the mean ± SEM of $n = 4$ independent experiments. **c** GST pull-down assay analyzing the effect of BH3-peptides on the interactions of GST-p53-DBD with His-BCL-2. The BH3 peptides are used as indicated. **d, e** GST pull-down assay analyzing the effect of p53-DBD on the interactions of GST-BCL-2 with Bax protein (**d**) or Bak protein (**e**). The source data are provided as a Source Data file.

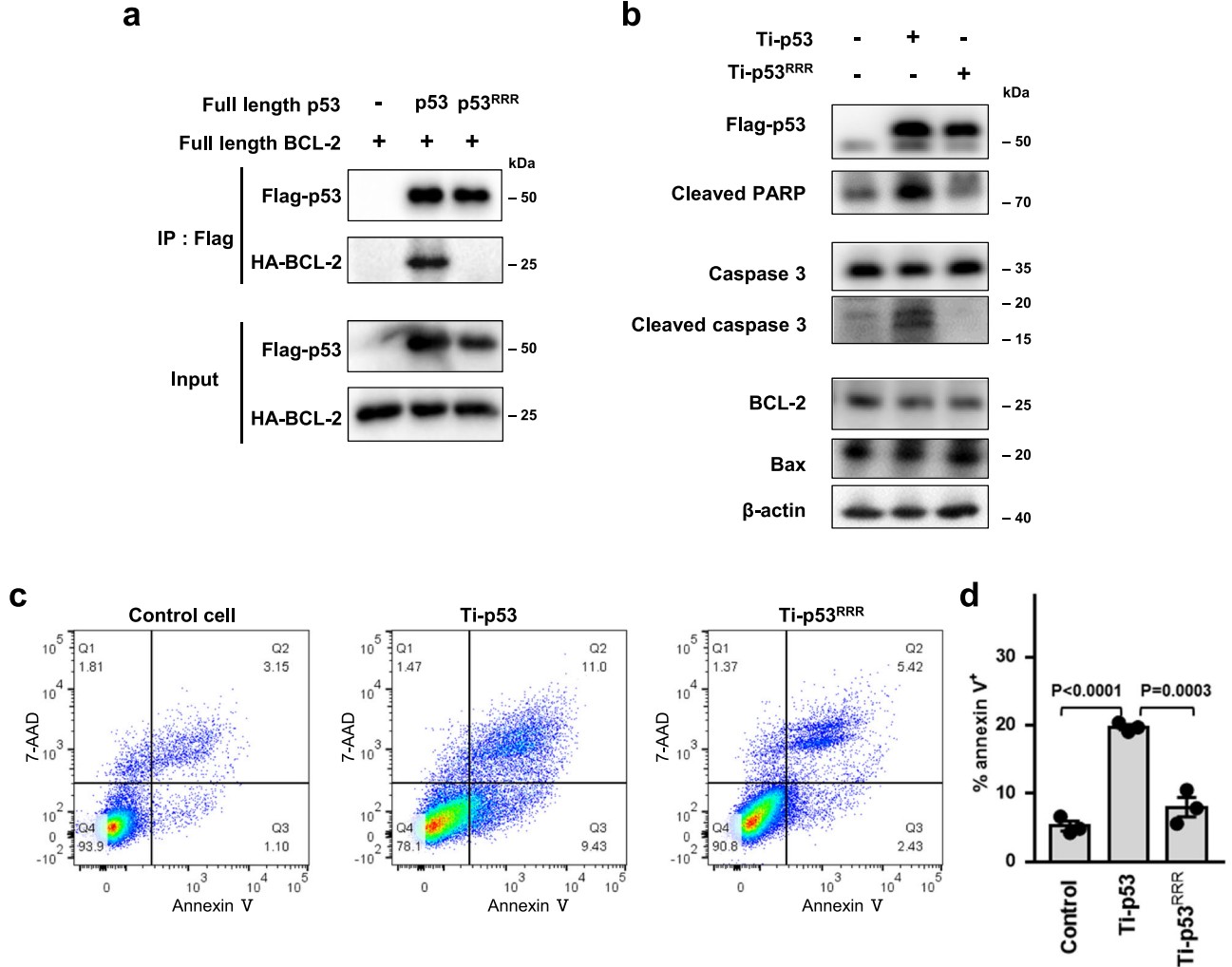

**Fig. 6 | Mutations at the p53/BCL-2 interface decrease p53-mediated apoptosis.**
**a** Coimmunoprecipitation determines the interaction between full-length p53 and full-length BCL-2. Full-length wild-type or mutant p53 plasmid (Flag-tagged) was co-transfected with full-length BCL-2 plasmid (HA-tagged) into HEK 293 T cells. Cell lysates were immunoprecipitated using anti-Flag beads and subjected to western blotting as indicated. **b** Caspase 3 and PARP activation detected by western blotting. p53$^{-/-}$ HCT116 cells stably expressing Ti-p53 or Ti-p53$^{RRR}$ were seeded in 6-well plates. After 48 h, cell lysates were immunoblotted with antibodies as indicated. **c**, **d** Flow cytometry detecting the apoptosis rate. p53$^{-/-}$ HCT116 cells stably expressing Ti-p53 or Ti-p53$^{RRR}$ were seeded in 6-well plates. After 48 h, the cells were stained with Annexin V-PE/7-AAD and subjected to flow cytometry. One representative experiment is shown in (**c**), as three independent replicates were similar. Data are represented as the mean ± SEM of $n = 3$ independent experiments (**d**). $P$ values were determined by one-way ANOVA followed by Dunnett's multiple comparisons test. Source data are provided as a Source Data file.

also reported that the association of p53 with BCL-xL releases Bax or recombinant activated Bid which was previously bound to BCL-xL[9]. The p53/BCL-xL complex was only disrupted by Puma but not by other pro-apoptotic BCL-2 family proteins[37]. Puma can induce partial unfolding of BCL-xL and disrupt the p53/BCL-xL interaction[38]. Nevertheless, the binding site on BCL-xL for p53 does not overlap with the BH3-binding pocket of BCL-xL[19–21]. To our surprise, the p53-DBD/BCL-2 structures presented here and the previous p53-DBD/BCL-xL structure[21] indicate that BCL-2 and BCL-xL had different interaction modes with p53 (Supplementary Figs. 9–10). In the crystallization and structure determination, we used glycine-rich linkers to help stabilize the interaction of p53-DBD with BCL-2 or BCL-xL, whereas the two complexes showed different patterns even when the same 16-residue linker was used. In addition, the BCL-2 residues at the BCL-2/p53-DBD interface or BCL-xL residues in the BCL-xL/p53-DBD interface are mostly conserved between these two proteins (Supplementary Figs. 9–10).

The reasons for the differences may come from several factors. One factor is the intrinsic difference between BCL-2 and BCL-xL.

Although BCL-2 and BCL-xL exert similar anti-apoptotic functions, the structural features of the highly conserved BH3 binding pocket dictate the sensitivity and specificity of BCL-xL and BCL-2 for different BH3 domains of pro-apoptotic proteins[27,39]. For example, BCL-2 preferentially binds to Bax, while BCL-xL preferentially binds to Bak[22,27]. Thus, the differences in the overall structure, including the conserved and less conserved regions, also result in different binding modes of BCL-xL and BCL-2 to p53. On the other hand, p53 is a highly dynamic protein that acts as a modular protein interacting with a wide range of DNAs and partner proteins. For example, p53-DBD also exhibits various interaction modes with ASPP family proteins ASPP2[40] and iASPP[41]. In terms of the interactions with BCL-2 and BCL-xL, p53 may indeed have different binding modes, or p53 may have two possible binding modes but the dominant mode is different. To understand these issues, more research is needed.

Mutations in the *TP53* gene and/or overexpression of BCL-2 help cancer cells evade apoptosis and develop drug resistance[42,43]. BH3-mimetic drugs have been developed to induce apoptosis by disrupting the interaction of BCL-2 with the BH3 domain of pro-apoptotic BCL-2

proteins[44]. The only FDA-approved BH3 mimetic is venetoclax (ABT199), which is a selective BCL-2 inhibitor. Recent studies have indicated that wild-type p53 activity contributes to apoptosis activation by BH3-mimetic drugs, whereas *TP53* mutations may confer resistance to BCL-2 targeted therapy[45–47]. It is unclear whether the direct interaction of p53 with BCL-2 has an effect on the activity of BH3-mimetic drugs, but it is noteworthy that both p53 and BH3-mimetic drugs bind in the BH3-binding pocket of BCL-2[28,48].

In our previous paper[21], based on the structure of the p53-DBD/BCL-xL complex and the sequence similarity between BCL-2 and BCL-xL, we predicted that BCL-2 might bind p53 in a manner similar to BCL-xL. Rather unexpectedly, our current study reveals a very different binding mode. Do BCL-2 and BCL-xL bind to p53 in different ways despite their high sequence similarity? Or do they bind to p53 through a dynamic binding mechanism? What does the full-length p53/BCL-2 complex look like in the cell? These questions need to be further addressed in the future. For crystallography, we used a truncation strategy similar to previous biochemical and structural studies of the BCL-2 family[25–28,49,50]. Several studies have reported that post-translational modifications or structural flexibility of the α1-α2 loop of BCL-2 or BCL-xL affect ligand access to the BH3 binding pocket[35,51–54]. It is possible that the α1-α2 loop of BCL-2 also plays a regulatory role in intracellular p53/BCL-2 interactions. It is also acknowledged that crystallography could introduce non-native crystal contact artefacts.

In summary, our study elucidates the structural basis of the p53/BCL-2 interaction and reveals the molecular mechanism by which p53 antagonizes BCL-2 activity by directly occupying the BH3-binding pocket and releasing pro-apoptotic BCL-2 family proteins sequestered in the pocket. These structural and functional data provide insight into the complicated regulatory mechanism of p53-mediated mitochondrial apoptosis.

## Methods
### Plasmids
Human BCL-2 (BCL-2, amino acids 1-205) was cloned into pGEx-4T1 (GE healthcare, 28-9545-49) with an N-terminal GST-tag and pET-28a (Novagen, 69864-3) with a C-terminal 6 × HIS-tag, respectively. The DNA sequences[25] of BCL-2#3 for crystallization were optimized and synthesized by GenScript. BCL-2#3 was cloned into pET-28a with a C-terminal 6 × HIS-tag. Human p53-DBD (amino acids 92-292) was cloned into pGEx-6P1 (GE healthcare, 28-9546-48). Human Bax (amino acids 1-171) was cloned into pGEx-6P1, and Bak (amino acids 22-186) was cloned into a modified pET-28a[55] with an N-terminal 6 × HIS-tag followed by a PreScission protease cleavage site. A BCL-2 construct connected to p53-DBD via a glycine-rich linker was PCR amplified and cloned into a modified pET-28a vector with an N-terminal 6 × HIS-tag.

Full-length p53 used in cellular assays was cloned into the retroviral vector pQCXIH (Clontech, 631516) and lentiviral vector pCDH-CMV-MCS-EF1-copGFP-T2A-Puro (System Biosciences, CD513B-1) with a C-terminal Flag-tag. Full-length BCL-2 was cloned into pQCXIH and pCDH-CMV-MCS-EF1-copGFP-T2A-Puro with an N-terminal HA-tag. To carry out the flow cytometry assay, the sequence of copGFP in the pCDH plasmids was deleted via the KOD-Plus-Mutagenesis Kit (TOYOBO, SMK-101).

All site-directed mutants were constructed using the wild-type plasmid as the template. All plasmids were performed according to the manufacturer's instructions for the ClonExpress II One Step Cloning Kit (Vazyme, C112). All plasmids were confirmed by DNA sequencing (Tsingke). The sequences of the primers are listed in Supplementary Table 3.

### Protein expression and purification
Proteins were recombinantly expressed in *E. coli* BL21 (DE3) cells (Transgen, CD601-02). The cells were grown at 37 °C in LB culture medium until the OD 600 reached 0.8. Then the cells were induced by 0.5 mM isopropyl β-D-1-thiogalactoside (IPTG) and further cultured at 18 °C for 8–12 h. Cells were harvested by centrifugation at 3000 × *g* for 20 min. For purification, cells were resuspended and lysed with a high-pressure homogenizer. The lysis buffer for BCL-2 family proteins was 20 mM Tris-HCl pH 8.0 and 500 mM NaCl. The lysis buffer for p53 proteins was 20 mM sodium citrate pH 6.3, 500 mM NaCl, 10 μM $Zn^{2+}$ and 3 mM β-ME. The cell lysate was further clarified by centrifugation at 20,000 × *g* for 30 min. Then, the supernatant was subjected to purification.

His-tagged BCL-2, BCL-2#3 and Bak proteins were purified by nickel affinity chromatography (GE Healthcare, 17-5318-02), followed by anion-exchange chromatography (Mono Q 5/50GL, GE Healthcare, 17-5166-01). Untagged Bak was obtained by incubation with PreScission protease at 4 °C before Mono Q. GST fusion proteins of BCL-2, p53-DBD and Bax were first purified by glutathione sepharose (GE Healthcare, 17-0756-01). Untagged p53-DBD and Bax were obtained by incubation with PreScission protease at 4 °C and then further purified by size exclusion chromatography Superdex 75 10/300 GL (GE Healthcare, 17-5174-01). For crystallization, the fused BCL-2-p53-DBD proteins were purified by nickel affinity chromatography. The 6 × HIS-tag was removed by PreScission protease at 4 °C. Then, the fusion proteins were further purified by cation-exchange chromatography (Mono S 5/50GL, GE Healthcare, 17-5168-01) and Superdex 75 10/300 GL.

### Crystallization
The fusion proteins at a concentration of approximately 10 mg/ml were used for initial crystal screening. Crystal screening was established by sitting drop vapour diffusion in an MRC 96-well double-drop plate by mixing 0.25 μl of protein solution and 0.25 μl of mother liquor. The mixture was equilibrated against 60 μl of the reservoir at 4 °C. Crystals appeared in some drops after 24–48 h, and then optimizations were carried out by hanging drop vapour diffusion in a 24-well plate by mixing 2.5 μl of protein solution (13 mg/ml) and 2.5 μl of reservoir solution against 600 μl of the reservoir at 4 °C. Final crystals of 16 residue-linked BCL-2-p53-DBD appeared in reservoir buffer containing 0.1 M HEPES:NaOH, pH 7.5, 20% (w/v) PEG 8000. Crystals of 19 and 22 residue-linked BCL-2-p53-DBD appeared in reservoir buffer containing 0.1 M imidazole pH 7.5–8.0, 15–21% (w/v) PEG 8000. For data collection, a single crystal was equilibrated in cryoprotectant consisting of reservoir solution supplemented with 20% (v/v) ethylene glycol and then flash-frozen in liquid nitrogen.

### Data collection and structure determination
Data were collected at beamlines BL17U[56] and BL19U1[57] of the Shanghai Synchrotron Radiation Facility (SSRF). HKL3000[58] was used to process the data. Structures were determined by molecular replacement (MR) using Phenix.phaser[59]. The BCL-2 structure (PDB: 6QG8)[25] and p53 structure (PDB: 3KMD)[60] were used as search templates. The initial model was manually built by the program Coot[61] and refinement was performed with Phenix.refine. Translational-liberation-screw (TLS) refinement was used during the last stages of refinement. Graphical representations of the structure were generated using PyMOL.

### Pull-down
Ten microliters of 10 μM GST-BCL-2 or His-BCL-2 was incubated with p53-DBD in 100 μL buffer for 2 h at 4 °C at a molar ratio of 1:1. Then glutathione-Sepharose or nickel beads were added to the mixture at 4 °C for 10 min. The samples were centrifuged at 300 × *g* for 5 min at 4 °C in a microcentrifuge. Beads were further washed 3–5 times. Finally, 1 × SDS-PAGE loading buffer was added to the beads. The samples were boiled in a water bath for 5 min and analyzed by western blotting. Buffer consisting of 1 × PBS supplemented with 0.5 mM TCEP

and 0.5% Triton X100 was used for GST pull-down. The same buffer with an additional 50 mM imidazole was used for Ni-NTA pull-down.

## Competition pull-down assay

To test the effects of pro-apoptotic BH3 peptides, 1 μM GST-p53-DBD was incubated with 1 μM HIS-BCL-2 and 5 μM BH3 peptide in 100 μL PBS buffer containing 0.5 mM TCEP. After incubation for 2 h, the samples were subjected to glutathione-Sepharose beads and incubated at 4 °C for 10 min. The samples were centrifuged at $300 \times g$ for 5 min at 4 °C in a microcentrifuge. Beads were further washed 3–5 times using PBS buffer containing 0.5 mM TCEP and 1% CHAPS. Finally, 100 μL 1 × SDS-PAGE loading buffer was added to the beads. All peptides were synthesized by GenScript with a purity over 98% as analyzed by HPLC, and dissolved in ddH$_2$O. The sequences of the peptides are listed in Supplementary Table 2.

To test the effects of pro-apoptotic proteins, 1 μM GST-BCL-2 was incubated with 1 μM Bax or Bak proteins in the absence or presence of p53-DBD proteins (2× gradient dilution, final concentration from 10 μM to 1.25 μM) in 100 μL PBS buffer containing 0.5 mM TCEP. After incubation at 4 °C overnight, samples were centrifuged at $15000 \times g$ for 5 min. The supernatant was subjected to the above GST pull-down assay using PBS buffer containing 0.5 mM TCEP and 1% CHAPS. The detergent CHAPS alone was reported not to affect the Bax or Bak structure[62,63]. Finally, 100 μL 1 × SDS-PAGE loading buffer was added to the beads. The samples were boiled in a water bath for 5 min and analyzed by western blotting.

## Enzyme linked immunosorbent assay (ELISA)

Protein samples were prepared according to the protocol described in Abcam's GST 6 × His-tag ELISA kit guide (Abcam, ab128573). Fifty microliters of His-BCL-2 protein at a concentration of 30 nM was added to the 6 × His-tag antibody pre-coated wells and incubated at 4 °C for 2 h. Fifty microliters of GST-p53-DBD at a gradient dilution concentration (final concentration from 1 nM to 30 μM) was incubated with wells after three washes. Then, the primary detector antibody and HRP-labelled secondary detector antibody were used accordingly. After the addition of HRP development solution for 15 min, a subsequent stop solution was added and the results were read at 450 nm. All assays were performed 4 times, and EC50 values were determined from a nonlinear dose-response model in GraphPad Prism 8.0.

## Microscale thermophoresis (MST)

MST binding assays were performed on a Monolith NT.115 instrument (NanoTemper Technology). Purified 6 × HIS tagged BCL-2 or BCL-2#3 was labelled with RED-Tris-NTA 2nd Generation dye (NanoTemper Technology, MO-L018) according to the manufacturer's instructions. GST-tagged p53-DBD at a concentration of 60 μM was diluted with PBS buffer (supplemented with 50 mM NaCl and 0.2 mM TCEP) in a 2:1 volume ratio. Then, p53-DBD was mixed with 100 nM labelled BCL-2 at room temperature in a 1:1 volume ratio. Thermophoresis data were recorded in expert mode with default parameters provided by MO. Control program. Fitting curves and KD values were generated by MO. Affinity Assay software.

## Fluorescence polarization assay (FPA)

Fluorescein-labelled Bax peptide (FITC-QDASTKKLSECLKRIGDELDS) was synthesized by GenScript with a purity over 95% as analyzed by HPLC and dissolved in ddH$_2$O. FPA was performed on an Enision multilabel reader (Perkin Elmer) using 96-well opti-plates (Corning, 4514). The binding reaction was carried out in 1× PBS with serial dilutions of His-BCL-2 (final concentration from 0.1 nM to 10 μM) and 10 nM FITC-labelled Bax peptide in a final volume of 150 μL. After 15 min of incubation at room temperature, the polarization in milli-polarization units (mP) was measured at room temperature with an excitation wavelength of 485 nm and an emission wavelength of 535 nm. All assays were performed 4 times, and EC50 values were determined from a nonlinear dose-response model in GraphPad Prism 8.0.

## Competition FPA

For the inhibition assay, fluorescence polarization was performed under the following conditions: each 96-well contains 15 nM fluorescent Bax-BH3 peptide, 20 nM His-BCL-2 protein and p53-DBD of serial dilutions (final concentration from 1 nM to 30 μM) in a final volume of 150 μL. After 15 min of incubation at room temperature, mP was measured as described above. For each assay, 15 nM free fluorescent Bax-BH3 peptide (namely fBax) and 20 nM BCL-2-bound fluorescent Bax-BH3 peptide in the absence of p53-DBD (namely mP$_0$) were used as controls on each assay plate. The percentage of inhibition was analyzed as previously[64]: $100[1-(mP-fBax)/(mP_0-fBax)]$. All assays were performed 4 times, and IC50 values were determined from a nonlinear dose-response model in GraphPad Prism 8.0.

## Cell lines and cell culture

HEK 293 T cells (American Type Culture Collection, CRL-3216) were cultured in DMEM/high glucose medium (Gibco, CT11995500BT) and p53$^{-/-}$ HCT 116 cells (Ubigene, YKO-H175) were cultured in McCoy's 5 A medium (Gibco, 16600082), all supplemented with 10% fetal bovine serum (Gibco, 10091148) and 1% penicillin and streptomycin (Gibco, 10091148) at 37 °C in a humidified incubator containing 5% CO$_2$.

## Stable cell line construction

To produce lentiviral particles, HEK 293 T cells were transfected with the packaging vectors pMD2.G (Addgene, 12259), psPAX2 (Addgene, 12260) and PCDH plasmids encoding Flag-tagged Ti-p53 or Ti-p53$^{RRR}$ as indicated. Viral supernatants were collected 48 and 72 h after transfection. Then, p53$^{-/-}$ HCT 116 cells were infected with 5 μg/mL polybrene and lentiviral particles. Cells were subjected to puromycin (2 μg/ml) screening for 3 days. Surviving cells were cultured in antibiotic-free medium for subsequent experiments. The transfection efficiency was verified by western blotting.

## Small interfering RNA transfections

HCT 116 cells stably expressing Ti-p53 were inoculated in 2 mL of antibiotic-free medium supplemented with 10% fetal bovine serum in 6-well culture plates. When cells reached 60–80% confluence, cells were transfected with jetPRIME (Polyplus, 101000046) for non-targeted small interfering RNA (siRNA) (negative control) or siRNA against specific human genes. After 48 h of transfection, cells were used for further experiments. All siRNAs were synthesized and purchased from Tsingke. Negative control siRNA was used, which does not correspond to any sequence in the human genome. The sequences are listed in Supplementary Table 4.

## Immunoprecipitation

HEK 293 T cells were cultured in 10 cm$^2$ dishes for 48 h and transiently transfected with pQCXIH-BCL-2 and pQCXIH-p53 plasmids as indicated using Lipofectamine 2000 (Invitrogen, 11668027) according to the manufacturer's protocol. After another 48 h, the cells were washed once with 5 ml PBS and lysed in 1 mL pre-chilled NP40 buffer (20 mM Tris-HCl pH 7.5, 150 mM NaCl, 0.3% NP-40, 1 mM PMSF) plus protease inhibitor cocktail. Lysates were incubated for 30 min at 4 °C. After centrifugation at $12000 \times g$ for 10 min at 4 °C, the lysates were subjected to immunoprecipitation with anti-Flag agarose beads (Sino Biological, 101274-MM13-RN) overnight at 4 °C. Beads were collected and washed three times with 1 mL of NP40 buffer. The complexes were eluted with 1× SDS-PAGE loading buffer for 10 min at 100 °C and then subjected to western blotting.

## Cell apoptosis analysis

HCT116 p53$^{-/-}$ cells stably expressing Ti-p53 or Ti-p53$^{RRR}$ and control cells were inoculated onto 6-well plates to ensure cell confluence of 30–50%. After 48 h, cell lysates were collected and western blotting was performed as indicated.

HCT116 p53$^{-/-}$ cells stably expressing Ti-p53 or Ti-p53$^{RRR}$ and control cells were cultured as described above and transiently transfected with the indicated siRNAs. Cells were washed with PBS and resuspended in 100 μl of binding buffer. Then, the cells were treated with an Annexin V-PE and 7AAD staining kit (Vazyme, A213) for 10–15 min in the dark at room temperature and subjected to flow cytometry analysis. FlowJo software was utilized to calculate the apoptotic cell rate.

## Western blotting analysis

Experiments were conducted using the following primary antibodies: Flag-tag antibody (Cell Signaling Technology, 8146 T, 1:1000 dilution), HA-tag antibody (Proteintech, 66006, 1:5000 dilution), caspase 3 antibody (Cell Signaling Technology, 9662 S, 1:1000 dilution), cleaved caspase 3 antibody (Cell Signaling Technology, 9661 S, 1:1000 dilution), cleaved PARP antibody (Cell Signaling Technology, 5625 S, 1:1000 dilution), β-actin antibody (Proteintech, 66009-1-Ig, 1:5000 dilution), Bax antibody (Cell Signaling Technology, 2772 S, 1:1000 dilution), BCL-2 antibody (Cell Signaling Technology, 15071, 1:1000 dilution), p53 antibody (Abcam, ab26, 1:1000 dilution) for p53-DBD, GST tag antibody (Cell Signaling Technology, 2625 S, 1:1000 dilution), and His-tag antibody (Abbkine, ABT2050, 1:5000 dilution). The secondary antibodies HRP-conjugated goat anti-mouse IgG (Abbkine, A21010, ATSDE1601, 1:2000 working dilution) and HRP-conjugated goat anti-rabbit IgG (Absin, abs20040, AS004, 1:2000 working dilution) were used. These antibodies were validated by western blotting according to the manufacturer's website. Bands were visualized by enhanced chemiluminescence detection reagents (Vazyme, E411-04). Full blots have been included in the Source data file.

## Statistics and reproducibility

For all immunoblotting experiments, three biologically independent replicates were performed with similar results, and a representative experiment is shown. For binding assays (MST, ELISA and FPA), all data are expressed as the mean ± SEM or mean ± SD for more than three independent experiments, and the exact number of replicates (n) is shown in the figure legend. For flow cytometry, three biological replicates were performed and one representative experiment is shown. Data were analyzed by one-way ANOVA followed by Dunnett's multiple comparisons test. Statistical analyses were performed using GraphPad Prism 8.0.

## Reporting summary

Further information on research design is available in the Nature Portfolio Reporting Summary linked to this article.

# Data availability

The coordinates and structure factors of the BCL-2/p53-DBD complex generated in this study have been deposited in the Protein Data Bank database under the accession codes 8HLL, 8HLM and 8HLN. Other data generated in this study are provided in the article and in the Supplementary Information/Source data file. The structure models used in the study are available in the Protein Data Bank under the following accession codes: 2OCJ (structure of apo p53-DBD), 1GJH (apo NMR solution structure of BCL-2), 2XA0 (structure of BCL-2/Bax-BH3 complex), 6QG8 (structure of BCL-2/Puma-BH3 complex), 3KMD (structure of p53/DNA complex) and 6LHD (BCL-xL/p53-DBD structure). A reporting summary is available as a Supplementary Information file. Source data are provided with this paper.

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

## Acknowledgements

The present study was financially supported by the Natural Science Foundation of China (grants 82273496 and 31900880 to H. Wei, 819740748 and 82172654 to Y.C.), Hunan Provincial Science and Technology Department (2018RS3026 and 2021RC4012 to Y.C.), China Postdoctoral Science Foundation (2019M652805 to H. Wei), Natural Science Foundation of Hunan Province (grants 2023JJ20092 to H. Wei, 2021JJ40961 to X.C., 2023JJ30863 to M.G.) and Central South University Innovation-Driven Research Program (2023CXQD076 to H. Wei). We thank the staff from the BL02U(former BL17U)/BL19U/BL10U beamlines of the National Facility for Protein Science in Shanghai (NFPS) at the Shanghai Synchrotron Radiation Facility for assistance during data collection. We thank Dr. Michael R. Stallcup (University of Southern California) for proofreading and suggestions.

## Author contributions

H. Wei, H. Wang, G.W., L.Q., L.J., X.C., Y.Z. and M.G. performed experiments; H. Wang, M.G., and S.D. performed data collection and structure determination. H. Wei, Z.C., Y.L., M.G. and Y.C. analyzed the data. H. Wei, H. Wang, M.G. and Y.C. prepared the manuscript.

## Competing interests

The authors declare no competing interests.
