## [Peer Review File · Nature Communications]

REVIEWER COMMENTS

Reviewer #1 (Remarks to the Author):

- What are the noteworthy results?

A crystal structure of the DNA binding domain (DBD) of p53 and a modified fragment of Bcl-2 has been reported. Key residues have been identified, purportedly revealing the p53-Bcl-2 interface (interacting residues).

- Will the work be of significance to the field and related fields? How does it compare to the established literature? If the work is not original, please provide relevant references.

Solving this structure of the interaction between p53-DBD and Bcl-2 is potentially highly significant, due to p53's role in blocking Bcl-2's anti-apoptotic activity. Solving the structure gives key information on the mechanism of action of p53's mitochondrial apoptosis pathway and how it disrupts Bcl-2 action by binding. This work appears to be original.

- Does the work support the conclusions and claims, or is additional evidence needed?

The critical issue with this structure is the use of a modified Bcl-2 (called Bcl-2#3) which lacks the flexible loop region of Bcl-2 (aa 32-68) which has been reported to interact with the DBD of p53 (Deng et al., Mol Cell Bio, 2006). Based on the reference to this Bcl-2#3 construct within this paper (and references therein), Bcl-2#3, which is used in this paper, instead has the first 34 aa's of Bcl-2, followed by the flexible domain from another protein, Bcl-XL (aas 35-50), then the Bcl-2 structure continues starting at residue 92. As shown in the diagram below (pasted from Petros et al., PNAS 2001, <https://doi.org/10.1073/pnas.041619798>)- see supplemental file- the flexible linker aa sequence (from around 32-68) from Bcl-2 (any of the 3 isoforms shown below) is vastly different from the flexible linker sequence from Bcl-XL.

Understandably, the authors used a modified Bcl-2 sequence due to the inability to crystallize native or truncated Bcl-2. Using Bcl-2#3 led to an "ability to achieve a high yield and was suitable for crystallization." Surprisingly, the details on this critical construct were not clearly explained in the paper.

Modifications in a protein to make it crystallize should ensure that the interacting domains between 2 proteins in question are intact. The substitution of the Bcl-XL flexible domain for the Bcl-2 flexible domain may not seem reasonable, given that p53-DBD binds to Bcl-2 via Bcl-2's original flexible domain, according to prior literature, which is no longer present in Bcl-2#3. The substituted flexible domain has a different length compared to Bcl-2's flexible domain, which may change the overall structure of Bcl-2 (and subsequently impact crystal packing) Unless the authors can rationalize/justify how this

substitution can be made without disrupting p53-DBD binding to Bcl-2, the basis of this paper seems flawed.

- Are there any flaws in the data analysis, interpretation and conclusions? - Do these prohibit publication or require revision?

The analysis, interpretation, and conclusions may be correct for the Bcl-2#3 interaction with p53-DBD, but due to the uncertainty of the validity of using Bcl-2#3 as a surrogate for Bcl-2, it is unclear if this represents a true Bcl-2/p53 interaction.

While for crystallography the Bcl-2#3 construct “had to be used” but for cell based assays, why wasn’t full length p53 also tested? Is there a loss of affinity of Bcl-2 to full length p53 vs. p53-DBD? Paper requires revision.

- Is the methodology sound? Does the work meet the expected standards in your field?

Most methodology is sound, well-described, and within the standards of the field. The authors are experts in the field of crystallography.

- Is there enough detail provided in the methods for the work to be reproduced?

Yes

Line 81- trial not trail

Line 249- mimetic not mimetics; also grammar issue with that sentence

Line 252- has not have

Line 258- grammar

Line 333- lowercase p53

Line 572- hot pink not hotpink

Reviewer #2 (Remarks to the Author):

Wei et al describe crystal structures of a complex between p53 and Bcl-2, together with functional data characterizing the interaction. Overall this is a very significant report that sheds light on an important interaction controlling apoptosis. The reported experiments are of high quality, and in their totality make for an exciting study. Data largely support conclusions.

Major issues:

1. The affinity measurements are based largely on ELISAs. The authors should conduct either SPR (Biacore) or ITC measurements to fully quantitate the interactions between p53 and Bcl-xL (and their respective mutants).
2. The data on apoptosis are all based on western blots. For these, full blots need to be shown. More importantly, the blots need to be supported by live cell analysis to show that there is a demonstrable effect on live cell apoptosis via FACS analysis and detection of an appropriate marker (PI release, PS via annexin etc). This is important to properly quantify the effect of mutations, and also allow cross-correlation to the significance of the p53-Bcl2 interaction vs p53-BclXL.
3. Considering the existence of the p53-BclXL complex, a detailed comparison of the two needs to be included.
4. Competition with Bax. This is an exciting finding. However, Bax is only one of the major pro-apoptotic Bcl-2 members that are able to interact with Bcl-2. In the absence of any other data on the remaining key pro-apoptotic regulators one cannot gauge as to how significant and relevant the Bax component is, and whether or not this is really driven by another pro-apoptotic protein that is being competed with. Another limitation is the reliance on Bax BH3 peptide (I note full length recombinant Bax is also used, but in the presence of triton – see comment further below on this). Whilst it is a very useful surrogate for Bax, to fully show that Bax is activated due to the p53 activity as per model and crystal structures this should be shown in a live cell system (ideally with and without Bax present). As is the suggestion of a significant role of Bax is insufficiently supported.

Minor issues:

1. For the reported apoptosis assays, is expression of Ti-p53 sufficient on its own to trigger apoptosis or was this supplemented by another stimulus?
2. Western blots need to be shown in full and loading controls need to be shown.
3. Number of experiments and error bars only defined in the legend for Fig 3D but not 3C or in Fig 6.
4. What concentration of protein was used for crystallization trials and to obtain the final crystals?
5. Triton X100 was present in the GST pull down buffer. Triton and certain other detergents are known to induce structural changes in Bax. Can the authors comment on the rationale for using Triton here in

light of this issue, and also comment on the possibility that triton may induce experimental artefacts in such a context and obfuscate the result.

6.

Typographical issues:

Line # - comment

43 – through forming formation of inhibitory complexes

47 – networks

51 – source? Nevertheless -> However?

53 –,(called the BH3-binding pocket).

60 – This Therefore, the

85 – formatting (spacing), To improve the success rate of crystallisation?,

88 – of all these three

92 – We hereafter henceforth used

93 – representative model?

94 – the p53-DBD

98,99 – wording

104 – corner regions.

111 – in vitro or in silico analysis?

123 – formed much less too informal/innacurate

129 – proteins

131 – before a few times followed by multiple

134 – these

140 – had an about approximate 5-fold decreased in affinity

147 – assays was were

148 – formed the prominent predominant

153 – at the p53/BCL-2

159 – Taking Taken together,

170 – They would thus clash and prevent each other This could lead to competition and prevent either protein

213 while however the detailed mechanistic interaction

214 – mechanism

215 – What's more However

233 – differences

244 – in cells -> in vitro?

249 – a wild-type p53 activity contributes

262 – activity through by directly

278 – missing tag description?

287 – sonication/centrifugation details?

296 – Then,

298 – GL.

324 – To carried carry

332,333 – consistency of values i.e., Fifty microliters. Consider rewording.

370 – Surviving cells are were

392 – was used.

Reviewer #3 (Remarks to the Author):

In article 405778 entitled “Structures of p53/BCL-2 complex suggest a mechanism for p53 to 1 antagonize BCL-2 activity” authors Wei, Wang et al. have determined three crystal structures of fusion proteins BCL-2-deltaTM–G-linker–p53DBD where they vary the linker size. All structures reveal that p53DBD binds to the BH3-binding groove of BCL-2, this mode of binding being different from the outside of the groove binding mode observed for BCL-xL-deltaTM–G-linker–p53DBD, which these authors previously characterized, and another study also resolved by NMR. The authors then produce mutants at the binding interface in both partners and characterize their binding to respective WT partner to validate the interactions site. Additionally, they selected a triple RRR mutant in p53DBD, which exhibited altered activity in one of their binding assays and reconstituted it in cells KO for p53 in the context of a transcriptionally inactive mutations previously described showing that it did not induce the typical markers of apoptosis. The direct regulation of the mitochondrial pathway at the BCL-2 family protein by cytosolic p53 has been a controversial area of investigation in the field that has been difficult to prove with existing knowledge and available reagents. Studies in this area therefore need to be carefully conducted.

While the structures reveal how p53 DBD may interact with BCL-2 and seem to have adequate statistics and resolution, I find the functional probing preliminary and not rigorously executed.

Here are some specific examples of problem area:

1. The side chain of D186 does not participate in the water bridge interaction as shown in the electron density in SF3b, instead the D186 CO being implicated. Why would the authors select to replace this residue with D186R, which abolished the pull-down interactions with GST-BCL-2, when presumably the side chain does not participate in the interaction.
2. For such high EC50 values observed in ELISA assays and high IC50 values observed in competitive FP assays, why did the authors not use a direct binding assay to characterize the interactions as they have done in their previous paper?
3. The assays in Figure 2 and Figure 6 should be combined as it deals with characterizing the same interaction. It is unclear why the authors have not included the RRR p53DBD mutant in the latter Figure 6b.
4. Related to Figure 6c and 6d it is remarkable that BAX peptide cannot fully displace p53DBD from BCL-2 suggesting possibly that other interactions may hold the complex in place. I cannot find the concentrations in the Methods section used in this assay which only show the stoichiometric considerations. The authors need to include this and do a proper titration to dislodge the BCL-2-p53 complex with excess BAX BH3 or BAX protein. It does not seem that the interaction is titratable.
5. The apoptosis data in Figure 4b is very preliminary and needs to be better developed. Cell death assay should show extent to cell death. Additionally, the authors need to at least show that the constructs do not induced apoptosis in p53-/- bak-/- bax-/- cell lines to exclude an off target effect especially as others have shown cytosolic p53-/- mediated cell death in the absence of BAK and BAX (<https://doi.org/10.1016/j.cell.2012.05.014>)
6. The authors need to characterize the homogeneity of the fusions used for crystallography on gel filtration chromatography (are these fusions forming oligomers?). They also need to acknowledge that crystallography could result in crystal contact artifacts, which may be non-native. For instance, if the fusions were p53DBD-G-linker-BCL-2-deltaTM they may lead to totally different crystal contacts. Therefore, the rigor of the investigation needs to be unquestionable.

Minor:

7. Figure 5 can follow immediately after Figure 2.
8. The model in Figure 6e is confusing with the p53-BCL-2 complex inhibiting the BAX-BCL-2 complex not making sense

- What are the noteworthy results?

A crystal structure of the DNA binding domain (DBD) of p53 and a modified fragment of Bcl-2 has been reported. Key residues have been identified, purportedly revealing the p53-Bcl-2 interface (interacting residues).

- Will the work be of significance to the field and related fields? How does it compare to the established literature? If the work is not original, please provide relevant references.

Solving this structure of the interaction between p53-DBD and Bcl-2 is potentially highly significant, due to p53's role in blocking Bcl-2's anti-apoptotic activity. Solving the structure gives key information on the mechanism of action of p53's mitochondrial apoptosis pathway and how it disrupts Bcl-2 action by binding. This work appears to be original.

- Does the work support the conclusions and claims, or is additional evidence needed?

The critical issue with this structure is the use of a modified Bcl-2 (called Bcl-2#3) which lacks the flexible loop region of Bcl-2 (aa 32-68) which has been reported to interact with the DBD of p53 (Deng et al., Mol Cell Bio, 2006). Based on the reference to this Bcl-2#3 construct within this paper (and references therein), Bcl-2#3, which is used in this paper, instead has the first 34 aa's of Bcl-2, followed by the flexible domain from another protein, Bcl-XL (aas 35-50), then the Bcl-2 structure continues starting at residue 92. As shown in the diagram below (pasted from Petros et al., PNAS 2001, <https://doi.org/10.1073/pnas.041619798>) the flexible linker aa sequence (from around 32-68) from Bcl-2 (any of the 3 isoforms shown below) is vastly different from the flexible linker sequence from Bcl-XL.

		$\alpha 1$										
		0000000000000000										
Bcl-xL	MSMAMSQS	NRELVVDFLS	YKLSQKGYSW	SQFS	DVEENR	TEAPEGTESE	METPSAINGN	PSWHLADSPA	VNGATGHSSS			
Bcl-2 (1)	MAHAGRTGYD	NREIVMKYIH	YKLSQRGYEW	DAGD	VGAAPP	GAAPAPGIFS	SQPGHTPHPA	ASRDPVARTS	PLQTPAAPGA			
Bcl-2 (2)	MAHAGRTGYD	NREIVMKYIH	YKLSQRGYEW	DAGD	VGAAPP	GAAPAPGIFS	SQPGHTPHPA	ASRDPVARTS	PLQTPAAPGA			
Bcl-2 (3)	MAHAGRTGYD	NREIVMKYIH	YKLSQRGYEW	DAGD	VGAAPP	GAAPAPGIFS	SQPGHTPHPA	ASRDPVARTS	PLQTPAAPGA			
Bcl-2 (1)/Bcl-xL	MAHAGRTGYD	NREIVMKYIH	YKLSQRGYEW	DAGD	DVEENR	TEAPEGTESE						
Bcl-2 (2)/Bcl-xL	MAHAGRTGYD	NREIVMKYIH	YKLSQRGYEW	DAGD	DVEENR	TEAPEGTESE						
	1	10	20	30	40	50	60	70	80			

Understandably, the authors used a modified Bcl-2 sequence due to the inability to crystallize native or truncated Bcl-2. Using Bcl-2#3 led to an “ability to achieve a high yield and was suitable for crystallization.” Surprisingly, the details on this critical construct were not clearly explained in the paper.

Modifications in a protein to make it crystallize should ensure that the interacting domains between 2 proteins in question are intact. The substitution of the Bcl-XL flexible domain for the Bcl-2 flexible domain may not seem reasonable, given that p53-DBD binds to Bcl-2 via Bcl-2’s original flexible domain, according to prior literature, which is no longer present in Bcl-2#3. The substituted flexible domain has a different length compared to Bcl-2’s flexible domain, which may change the overall structure of Bcl-2 (and subsequently impact crystal packing) Unless the authors can rationalize/justify how this substitution can be made without disrupting p53-DBD binding to Bcl-2, the basis of this paper seems flawed.

- Are there any flaws in the data analysis, interpretation and conclusions? - Do these prohibit publication or require revision?

The analysis, interpretation, and conclusions may be correct for the Bcl-2#3 interaction with p53-DBD, but due to the uncertainty of the validity of using Bcl-2#3 as a surrogate for Bcl-2, it is unclear if this represents a true Bcl-2/p53 interaction.

While for crystallography the Bcl-2#3 construct “had to be used” but for cell based assays, why wasn’t full length p53 also tested? Is there a loss of affinity of Bcl-2 to full length p53 vs. p53-DBD? Paper requires revision.

- Is the methodology sound? Does the work meet the expected standards in your field?

Most methodology is sound, well-described, and within the standards of the field. The authors are experts in the field of crystallography.

- Is there enough detail provided in the methods for the work to be reproduced?

Yes

Line 81- trial not trail

Line 249- mimetic not mimetics; also grammar issue with that sentence

Line 252- has not have

Line 258- grammar

Line 333- lowercase p53

Line 572- hot pink not hotpink

Point-by-point response

We would like to express our gratitude to all the reviewers for their valuable and constructive comments on our manuscript. We are submitting a revised version of the manuscript, in which we have carefully addressed all the comments, incorporated additional experiments, and clarified the text. Below is a point-by-point response to the reviewers' comments.

REVIEWER COMMENTS

Reviewer #1 (Remarks to the Author):

1. The critical issue with this structure is the use of a modified Bcl-2 (called Bcl-2#3) which lacks the flexible loop region of Bcl-2 (aa 32-68) which has been reported to interact with the DBD of p53 (Deng et al., Mol Cell Bio, 2006). Based on the reference to this Bcl-2#3 construct within this paper (and references therein), Bcl-2#3, which is used in this paper, instead has the first 34 aa's of Bcl-2, followed by the flexible domain from another protein, Bcl-XL (aas 35-50), then the Bcl-2 structure continues starting at residue 92. As shown in the diagram below (pasted from Petros et al., PNAS 2001, <https://doi.org/10.1073/pnas.041619798>)- see supplemental file- the flexible linker aa sequence (from around 32-68) from Bcl-2 (any of the 3 isoforms shown below) is vastly different from the flexible linker sequence from Bcl-XL.

Understandably, the authors used a modified Bcl-2 sequence due to the inability to crystallize native or truncated Bcl-2. Using Bcl-2#3 led to an "ability to achieve a high yield and was suitable for crystallization." Surprisingly, the details on this critical construct were not clearly explained in the paper.

Modifications in a protein to make it crystallize should ensure that the interacting domains between 2 proteins in question are intact. The substitution of the Bcl-XL flexible domain for the Bcl-2 flexible domain may not seem reasonable, given that p53-DBD binds to Bcl-2 via Bcl-2's original flexible domain, according to prior literature, which is no longer present in Bcl-2#3. The substituted flexible domain has a different length compared to Bcl-2's flexible domain, which may change the overall structure of Bcl-2 (and subsequently impact crystal packing) Unless the authors can rationalize/justify how this substitution can be made without disrupting p53-DBD binding to Bcl-2, the basis of this paper seems flawed.

Response: We thank the reviewer for bringing up the concern. Based on the reviewer's suggestion, the sequences of BCL-2 constructs have been provided in the revised Supplementary Figure 1, and the details on the critical constructs have been explained in the revised manuscript. BCL-2#3 is a truncated BCL-2 used in a previous structural study, featuring a shortened $\alpha 1$ - $\alpha 2$ loop and certain surface residues replaced with a BCL-xL sequence (ref 25). BCL-2#3 has been reported to have a similar selectivity profile for BH3-only peptides as wild-type BCL-2 (ref 25). A similar truncated strategy has been employed in many biochemical and structural studies of the BCL-2 family (ref 25-30).

To investigate whether the modifications affect the interaction between BCL-2 and p53-DBD, we carried out MST to determine the binding affinity. The results showed that p53-DBD had similar Kd values for both BCL-2 (3.5 μ M) and BCL-2#3 (3.9 μ M), suggesting that the modifications of BCL-2 didn't influence its binding affinity to p53-DBD (revised Supplementary Figure 3). In the p53-DBD/BCL-2#3 structures, the flexible loop and modified residues are not located on the contact

surface with p53-DBD (revised Supplementary Figure 4c). The overall structure of BCL-2#3 closely resembles the apo solution structure of BCL-2 (PDB: 1GJH) (revised Supplementary Figure 4b) or the BCL-2 in the BCL-2/Bax-BH3 structure (PDB: 2XA0) (revised Figure 4b). This indicates the modification has minimal influence on the overall structure of BCL-2. In addition, mutational analysis based on wild-type recombinant BCL-2 (revised Figure 3) or full-length BCL-2 (revised Figure 6a) supports the interface defined by the p53-DBD/BCL-2#3 structures. Collectively, these data suggest that the modifications of BCL-2 do not disrupt the binding of p53-DBD to BCL-2.

In the study mentioned by the reviewer (Deng et al., Mol Cell Bio, 2006), cell-based co-immunoprecipitation demonstrated that a BCL-2 construct with a deletion of amino acids 32-68 failed to co-precipitated with p53 in H7 cells. Notably, several studies have reported that post-translational modifications or structural flexibility of the $\alpha 1$ - $\alpha 2$ loop (aa 32-92) of BCL-2 or BCL-xL can affect ligand access into the BH3 binding pocket (ref 36, 50-53). It is possible that the $\alpha 1$ - $\alpha 2$ loop of BCL-2 also plays a regulatory role in the intracellular p53/BCL-2 interactions. We have addressed the issue in the revised discussion section.

2. The analysis, interpretation, and conclusions may be correct for the Bcl-2#3 interaction with p53-DBD, but due to the uncertainty of the validity of using Bcl-2#3 as a surrogate for Bcl-2, it is unclear if this represents a true Bcl-2/p53 interaction.

While for crystallography the Bcl-2#3 construct “had to be used” but for cell based assays, why wasn’t full length p53 also tested? Is there a loss of affinity of Bcl-2 to full length p53 vs. p53-DBD? Paper requires revision.

Response: Thanks for the comment. In our study, BCL-2#3 was solely used for crystallography, and all the biochemical assays were conducted using recombinant wild-type BCL-2. For the cell-based assays, full-length p53 and full-length BCL-2 were used (revised Figure 6). To better distinguish these constructs, we have classified BCL-2 constructs as BCL-2#3, recombinant BCL-2 and FL-BCL-2, and p53 constructs as p53-DBD, FL-p53 and Ti-p53 (Ti-p53 is generated based on FL-p53) in the revised manuscript.

Line 81- trial not trail

Line 249- mimetic not mimetics; also grammar issue with that sentence

Line 252- has not have

Line 258- grammar

Line 333- lowercase p53

Line 572- hot pink not hotpink

Response: We appreciate the careful reading by the reviewer. We have made corresponding modifications to the wording and phrases in the revised manuscript.

Reviewer #2 (Remarks to the Author):

Wei et al describe crystal structures of a complex between p53 and Bcl-2, together with functional data characterizing the interaction. Overall, this is a very significant report that sheds light on an important interaction controlling apoptosis. The reported experiments are of high quality, and in their totality make for an exciting study. Data largely support conclusions.

Major issues:

1. The affinity measurements are based largely on ELISAs. The authors should conduct either SPR

(Biacore) or ITC measurements to fully quantitate the interactions between p53 and Bcl-xL (and their respective mutants).

Response: We thank the reviewer for the positive comments and the suggestion. We speculate that the issue raised here by the reviewer is about the interaction of p53 with BCL-2. In fact, we attempted Biacore X100 and ITC measurements; however, both recombinant p53-DBD and BCL-2 proteins, when present at high concentrations, tend to precipitate after prolonged exposure to the instruments at room temperature. In the revised manuscript, we employed MST to quantitate the interactions between p53-DBD and BCL-2. GST-p53-DBD was used to help stabilize the p53 protein. The GST tag was tested and did not bind to BCL-2. After the MST experiments, the samples remained transparent. The results are presented in the revised Supplementary 3 and the revised Figure 3c-d. The descriptions and detailed methods have been added to the revised manuscript. The data from both methods have shown that the structure-based mutations decreased the binding ability of p53-DBD to BCL-2, and supported the p53-DBD/BCL-2#3 interface defined by the complex structure. We have also observed that the K_d values of MST were different from the EC₅₀ values of ELISA (revised Supplementary 6), possibly due to the utilization of different methods.

2. The data on apoptosis are all based on western blots. For these, full blots need to be shown. More importantly, the blots need to be supported by live cell analysis to show that there is a demonstrable effect on live cell apoptosis via FACS analysis and detection of an appropriate marker (PI release, PS via annexin etc). This is important to properly quantify the effect of mutations, and also allow cross-correlation to the significance of the p53-Bcl2 interaction vs p53-BclXL.

Response: We appreciate the suggestions. Per the reviewer's suggestion, the full blots have been included in the revised Supplementary Figure 12. Additionally, we have used FACS analysis and detection of Annexin-V/7-AAD marker to demonstrate the effect of p53 mutations on live cell apoptosis (revised Figure 6c-d, revised Supplementary Figure 8). Ti-p53 induced approximately 19.6% of apoptotic cells, whereas the ratio of apoptotic cells reduced significantly to 8% with Ti-p53^{RRR}. The relevant descriptions have been added to the revised Results section.

3. Considering the existence of the p53-BclXL complex, a detailed comparison of the two needs to be included.

Response: Thanks for the suggestion. We have provided a detailed comparison of p53/BCL-2 complex with p53/BCL-xL complex in the revised Supplementary Figure 10-11.

4. Competition with Bax. This is an exciting finding. However, Bax is only one of the major pro-apoptotic Bcl-2 members that are able to interact with Bcl-2. In the absence of any other data on the remaining key pro-apoptotic regulators one cannot gauge as to how significant and relevant the Bax component is, and whether or not this is really driven by another pro-apoptotic protein that is being competed with. Another limitation is the reliance on Bax BH3 peptide (I note full length recombinant Bax is also used, but in the presence of triton – see comment further below on this). Whilst it is a very useful surrogate for Bax, to fully show that Bax is activated due to the p53 activity as per model and crystal structures this should be shown in a live cell system (ideally with and without Bax present). As is the suggestion of a significant role of Bax is insufficiently supported.

Response: We thank the reviewer for raising these concerns. In the revised manuscript, we have modified the GST pull-down assay by replacing triton with a buffer containing CHAPS, which has

been reported not to alter the Bax or Bak conformation (ref 60-61). We utilized GST pull-down to detect the effects of p53-DBD on the interactions of BCL-2 with BH3 peptides (Bid, Bim, Puma, Bax, and Bak) (revised Figure 5c), as well as Bax proteins (revised Figure 5d), and Bak proteins (revised Figure 5e). The results indicate a negative correlation between the complex formation of p53-DBD/BCL-2 and the complex formation of BCL-2 with pro-apoptotic peptides or proteins. The descriptions of these findings have been added to the revised Results section.

To examine whether Bax or Bak is activated by Ti-p53 activity in a cell system, we knocked down Bax or Bak by small interfering RNA (siRNA) in Ti-p53 stably-expressing HCT116 cells, and then used FACS analysis and detection of an Annexin-V/7-AAD marker to detect cell apoptosis (revised Supplementary Figure 9). Compared to Ti-p53 (19.6%), Bax siRNA and Bak siRNA decreased the apoptotic cell to 12.1% and 13.5%, respectively. The simultaneous knockdown of Bax and Bak still led to 9.6% of cell apoptosis. The results indicate that apoptosis induced by Ti-p53 is dependent on both Bax and Bak. The related descriptions have been added to the revised Results section.

Minor issues:

1. For the reported apoptosis assays, is expression of Ti-p53 sufficient on its own to trigger apoptosis or was this supplemented by another stimulus?

Response: Yes. The expression of Ti-p53 is sufficient on its own to trigger apoptosis, and no additional stimulus is required. We have provided a detailed description of the experimental condition in the revised Methods section.

2. Western blots need to be shown in full and loading controls need to be shown.

Response: In the revised manuscript, western blots have been shown in full (revised Supplementary Figure 12). We have carefully examined all the loading controls. Please kindly bring it to our attention if any information appears to be missing.

3. Number of experiments and error bars only defined in the legend for Fig 3D but not 3C or in Fig 6.

Response: Thanks for the suggestion. We have accordingly added the number of experiments and error bars in the revised manuscript. For specific data points, the error bars may be shorter than the height of the symbol, making them less visually prominent.

4. What concentration of protein was used for crystallization trials and to obtain the final crystals?

Response: The proteins were used for initial crystal screening at approximately 10 mg/ml. Through optimization, final crystals were obtained using 13 mg/ml protein concentration. We have added detailed information to the revised Methods section in the revised manuscript.

5. Triton X100 was present in the GST pull down buffer. Triton and certain other detergents are known to induce structural changes in Bax. Can the authors comment on the rationale for using Triton here in light of this issue, and also comment on the possibility that triton may induce experimental artefacts in such a context and obfuscate the result.

Response: We thank the reviewer for the valuable comments. In the revised manuscript, we have modified the GST pull-down assay by replacing triton with a buffer containing CHAPS, which has

been reported not to alter the Bax or Bak conformation (ref 60-61). We then re-performed the experiments and the results are presented in the revised Figure 5c-e. The experimental procedure has been provided in the revised Methods section.

Typographical issues:

Line # - comment

43 – through forming formation of inhibitory complexes

47 – networks

51 – source? Nevertheless -> However?

53 –,(called the BH3-binding pocket).

60 – This Therefore, the

85 – formatting (spacing), To improve the success rate of crystallisation?,

88 – of all these three

92 – We hereafter henceforth used

93 – representative model?

94 – the p53-DBD

98,99 – wording

104 – corner regions.

111 – in vitro or in silico analysis?

123 – formed much less too informal/innacurate

129 – proteins

131 – before a few times followed by multiple

134 – these

140 – had an about approximate 5-fold decreased in affinity

147 – assays was were

148 – formed the prominent predominant

153 – at the p53/BCL-2

159 – Taking Taken together,

170 – They would thus clash and prevent each other This could lead to competition and prevent either protein

213 while however the detailed mechanistic interaction

214 – mechanism

215 – What's more However

233 – differences

244 – in cells -> in vitro?

249 – a wild-type p53 activity contributes

262 – activity through by directly

278 – missing tag description?

287 – sonication/centrifugation details?

296 – Then,

298 – GL.

324 – To carried carry

332,333 – consistency of values i.e., Fifty microliters. Consider rewording.

370 – Surviving cells are were

392 – was used.

Response: We appreciate the reviewer's careful reading and valuable feedback. We have addressed the typographical issues and revised the grammar and references accordingly.

Reviewer #3 (Remarks to the Author):

In article 405778 entitled “Structures of p53/BCL-2 complex suggest a mechanism for p53 to 1 antagonize BCL-2 activity” authors Wei, Wang et al. have determined three crystal structures of fusion proteins BCL-2-deltaTM-G-linker-p53DBD where they vary the linker size. All structures reveal that p53DBD binds to the BH3-binding groove of BCL-2, this mode of binding being different from the outside of the groove binding mode observed for BCL-xL-deltaTM-G-linker-p53DBD, which these authors previously characterized, and another study also resolved by NMR. The authors then produce mutants at the binding interface in both partners and characterize their binding to respective WT partner to validate the interactions site. Additionally, they selected a triple RRR mutant in p53DBD, which exhibited altered activity in one of their binding assays and reconstituted it in cells KO for p53 in the context of a transcriptionally inactive mutations previously described showing that it did not induce the typical markers of apoptosis. The direct regulation of the mitochondrial pathway at the BCL-2 family protein by cytosolic p53 has been a controversial area of investigation in the field that has been difficult to prove with existing knowledge and available reagents. Studies in this area therefore need to be carefully conducted.

While the structures reveal how p53 DBD may interact with BCL-2 and seem to have adequate statistics and resolution, I find the functional probing preliminary and not rigorously executed.

Here are some specific examples of problem area:

1. The side chain of D186 does not participate in the water bridge interaction as shown in the electron density in SF3b, instead the D186 CO being implicated. Why would the authors select to replace this residue with D186R, which abolished the pull-down interactions with GST-BCL-2, when presumably the side chain does not participate in the interaction.

Response: We thank the reviewer for pointing out the concern. In the structures determined in our study, main chain of D186 participate in the water-bridged interaction with BCL-2 residues R107 and Y108 in the structure of the 22-residue linked protein (revised Supplementary Figure 5a), while in the structures of the 16-residue or 19-residue linked proteins, the side chain of residue D186 is involved in the direct interactions with BCL-2 residues R107 or Y108 (revised Supplementary Figure 5b-c). We, therefore, hypothesized that D186 could play an important role for the p53-DBD/BCL-2 interactions, and designed mutation D186R.

2. For such high EC50 values observed in ELISA assays and high IC50 values observed in competitive FP assays, why did the authors not use a direct binding assay to characterize the interactions as they have done in their previous paper?

Response: Thanks for the comment. We attempted to characterize the interactions between p53-DBD and BCL-2 using Biacore X100 and ITC measurements. However, both recombinant p53-DBD and BCL-2 proteins, when present at high concentrations, tend to precipitate after prolonged exposure to the instruments at room temperature. In the revised manuscript, we employed MST method to characterize the direct binding ability. GST-p53-DBD was used to help stabilize the p53 proteins. The GST tag was tested and did not bind to the BCL-2 protein. After the MST experiments, the samples remained transparent. The results are presented in the revised Supplementary 3 and the

revised Figure 3c-d. The descriptions and detailed methods have been added to the revised manuscript. The data from both methods have shown that these structure-based mutations decreased the binding ability of p53-DBD to BCL-2, and supported the p53-DBD/BCL-2#3 interface defined by the complex structure. We have also observed that the K_d values of MST are different from the EC₅₀ values of ELISA (revised Supplementary 6), possibly due to the utilization of different methods.

3. The assays in Figure 2 and Figure 6 should be combined as it deals with characterizing the same interaction. It is unclear why the authors have not included the RRR p53DBD mutant in the latter Figure 6b.

Response: We guess the issue raised here by the reviewer refers to Figure 3 and Figure 6 in the original manuscript. According to the reviewer's suggestion, we have adjusted the figures in the revised manuscript. The revised figures 2-3 show the detailed interactions of p53/BCL-2 complex, and the revised figures 4-5 show the effect of p53/BCL-2 complex on the interaction of BCL-2 with pro-apoptotic proteins. The results of the cell-based assay are shown in the revised Figure 6.

Regarding the competitive FPA (revised Figure 5b), the p53-DBD mutant RRR at 30 μM failed to cause a 30% change in the fluorescence signal of BCL-2/fBAX. Due to such a low signal-to-noise ratio, we did not proceed with the assay to obtain the IC₅₀ value.

4. Related to Figure 6c and 6d it is remarkable that BAX peptide cannot fully displace p53DBD from BCL-2 suggesting possibly that other interactions may hold the complex in place. I cannot find the concentrations in the Methods section used in this assay which only show the stoichiometric considerations. The authors need to include this and do a proper titration to dislodge the BCL-2-p53 complex with excess BAX BH3 or BAX protein. It does not seem that the interaction is titratable.

Response: We appreciate the reviewer's comment. In the revised manuscript, we have adjusted the competitive GST pull-down assay according to the reviewers' suggestions. Detailed information on protein concentration has been provided in the revised Methods section. We have repeated the experiments with titration as suggested. A gradient dilution of p53-DBD was used to compete for BCL-2 with Bax (revised Figure 5d) or Bak proteins (revised Figure 5e). When the amount of p53-DBD is about 10-times higher than Bax (Figure 5d, lane 5) or Bak proteins (Figure 5e, lane 5), p53-DBD cannot fully displace Bax or Bak proteins from GST-BCL-2. We acknowledge that the interaction is not fully titratable, possibly due to other interactions that may hold the complex in place. We have included the information in the revised Results section.

5. The apoptosis data in Figure 4b is very preliminary and needs to be better developed. Cell death assay should show extent to cell death. Additionally, the authors need to at least show that the constructs do not induced apoptosis in p53^{-/-} bak^{-/-} bax^{-/-} cell lines to exclude an off target effect especially as others have shown cytosolic p53^{-/-} mediated cell death in the absence of BAK and BAX (<https://doi.org/10.1016/j.cell.2012.05.014>).

Response: We thank the reviewer for the comment. Based on the reviewer's suggestion, we have further developed the cell apoptosis assays (revised Figure 6, revised Supplementary Figure 8). The extent of cell death has been assessed by FACS analysis after staining with Annexin V-PE/7-AAD kit. To further analyze the target effect of Ti-p53, we knocked down Bax or/and Bak by small interfering RNA (siRNA) in Ti-p53 stably-expressing HCT116 cells, and then used FACS analysis

to detect cell apoptosis (revised Supplementary Figure 9). Compared to Ti-p53 (19.6%), Bax siRNA and Bak siRNA decreased the apoptotic cell to 12.1% and 13.5%, respectively. With the simultaneous knockdown of Bax and Bak, there was still a significant level of cell apoptosis at 9.6%. This data suggested that Ti-p53-mediated apoptosis in HCT116 cells depended on both Bax and Bak, and Ti-p53 may also have other targets. We have added the information in the revised Results section.

6. The authors need to characterize the homogeneity of the fusions used for crystallography on gel filtration chromatography (are these fusions forming oligomers?). They also need to acknowledge that crystallography could result in crystal contact artifacts, which may be non-native. For instance, if the fusions were p53DBD–G-linker–BCL-2-deltaTM they may lead to totally different crystal contacts. Therefore, the rigor of the investigation needs to be unquestionable.

Response: Gel filtration chromatography to characterize the homogeneity of the fusions used for crystallography has been added in the revised supplementary Figure 2. In the revised Discussion section, we have acknowledged that crystallography could introduce non-native crystal contact artifacts.

Minor:

7. Figure 5 can follow immediately after Figure 2.

Response: Thanks for the suggestions. We have adjusted the figures in the revised manuscript. The revised figures 2-3 present the detailed interactions of p53/BCL-2 complex, and the revised figures 4-5 illustrate the effect of p53/BCL-2 complex on the interaction of BCL-2 with pro-apoptotic proteins.

8. The model in Figure 6e is confusing with the p53–BCL-2 complex inhibiting the BAX–BCL-2 complex not making sense

Response: We have removed the model based on the reviewer's suggestion and additional data incorporated in the revised manuscript.

REVIEWERS' COMMENTS

Reviewer #1 (Remarks to the Author):

The authors have addressed my main scientific concerns adequately. The paper is much more clear now. A few comments:

Supplemental figure 1 seems so critical – should appear in the paper not as a supplement but as a figure in the paper.

There are so many grammar errors- some are corrected below, and they are mostly in the newly added sections.

85 used a fusion strategy. First, BCL-2 was linked to p53-DBD via a glycine-rich linker that was
86 previously used for p53-DBD/BCL-xL complex to increase the proximity between the two
87 proteins.²¹ However, we noted a BCL-2 construct (BCL-2#3), featuring a shortened $\alpha 1$ - $\alpha 2$
88 loop and certain surface residues replaced with a BCL-xL sequence (Supplementary Figure 1),
89 could be expressed in high yield and was suitable for crystallization²⁵.

98 BCL-2 did not influence

(do not use contractions in scientific writing)

195 Given that several pro-apoptotic BCL-2 members

231 assay and detection of an Annexin-V/7-AAD marker showed that Ti-p53 induced

232 approximately 19.6% of cells to undergo apoptosis, whereas the ratio of apoptotic cells reduced

239 p53 (19.6%), Bax siRNA and Bak siRNA decreased the apoptotic cells to 12.1% and 13.5%,

Line 252 The binding sites of p53 in BCL-2 largely overlap with those of pro-apoptotic members of the BCL-2 family, and thus p53 can competitively inhibit the interaction of BCL-2 with pro-apoptotic family members

Line 321 A BCL-2 construct connected to p53-DBD via a glycine rich linker was PCR amplified and cloned into a modified pET-28a...

Line 324 Full-length p53 used in cellular assays was cloned into the retroviral vector pQCXIH
325 (Clontech, 631516) and lentiviral vector pCDH-CMV-MCS-EF1-copGFP-T2A-Puro (System
326 Biosciences, CD513B-1) with a C-terminal Flag-tag. Full-length BCL-2 was cloned into
327 pQCXIH and pCDH-CMV-MCS-EF1-copGFP-T2A-Puro with a N-terminal HA-tag. To carry
328 out flow cytometry assay, the sequence of copGFP in the pCDH plasmids was deleted via
329 KOD -Plus-Mutagenesis Kit (TOYOBO, SMK-101).

Line 339 The lysis buffer for BCL-2 family proteins was 20 mM Tris-HCl pH 8.0, and 500 mM NaCl. The lysis buffer for p53 proteins was 20 mM sodium...

Line 346 Untagged Bak was obtained by incubation with PreScission protease at 4°C before Mono Q

Line 399 The detergent CHAPS alone was reported not to affect the Bax or Bak structure^{61,62}

Reviewer #2 (Remarks to the Author):

The authors have addressed all my concerns to my full satisfaction. Only a few minor issues remain:

Line 98 - BCL-2 did not influence its binding affinity

Line 149 – I think this should be EC50

Line 152 – did the authors determine Kd or KD with their assays?

Line 305 – artefacts

Line 500 – Protein structure coordinate availability

Figure 3: Comparison of Kd fit:

Figure 3: Mutational analysis

Line 720: Curves were fitted and Kd values of interactions calculated using experimental data with a Kd model.....

It would be good if the changes are applied to the same issues in the supplementary data (KD vs Kd, EC50 etc)

Reviewer #3 (Remarks to the Author):

The authors have addressed my comments through additional experimentation when necessary. Thank you

Point-by-point response

We are grateful to all reviewers for their valuable and positive comments. We are submitting a revised version of the manuscript in which we have carefully addressed all comments. The following is a point-by-point response to the reviewers' comments.

REVIEWERS' COMMENTS

Reviewer #1 (Remarks to the Author):

The authors have addressed my main scientific concerns adequately. The paper is much more clear now. A few comments:

Supplemental figure 1 seems so critical – should appear in the paper not as a supplement but as a figure in the paper.

Response: We thank the reviewer for the suggestion. We have moved the figure into the revised Figure 1a-b.

There are so many grammar errors- some are corrected below, and they are mostly in the newly added sections.

85 used a fusion strategy. First, BCL-2 was linked to p53-DBD via a glycine-rich linker that was 86 previously used for p53-DBD/BCL-xL complex to increase the proximity between the two 87 proteins.²¹ However, we noted a BCL-2 construct (BCL-2#3), featuring a shortened $\alpha 1$ - $\alpha 2$ 88 loop and certain surface residues replaced with a BCL-xL sequence (Supplementary Figure 1), 89 could be expressed in high yield and was suitable for crystallization²⁵.

98 BCL-2 did not influence
(do not use contractions in scientific writing)

195 Given that several pro-apoptotic BCL-2 members

231 assay and detection of an Annexin-V/7-AAD marker showed that Ti-p53 induced
232 approximately 19.6% of cells to undergo apoptosis, whereas the ratio of apoptotic cells reduced

239 p53 (19.6%), Bax siRNA and Bak siRNA decreased the apoptotic cells to 12.1% and 13.5%,

Line 252 The binding sites of p53 in BCL-2 largely overlap with those of pro-apoptotic members of the BCL-2 family, and thus p53 can competitively inhibit the interaction of BCL-2 with pro-apoptotic family members

Line 321 A BCL-2 construct connected to p53-DBD via a glycine rich linker was PCR amplified and cloned into a modified pET-28a...

Line 324 Full-length p53 used in cellular assays was cloned into the retroviral vector pQCXIH 325 (Clontech, 631516) and lentiviral vector pCDH-CMV-MCS-EF1-copGFP-T2A-Puro (System 326 Biosciences, CD513B-1) with a C-terminal Flag-tag. Full-length BCL-2 was cloned into

327 pQCXIH and pCDH-CMV-MCS-EF1-copGFP-T2A-Puro with a N-terminal HA-tag. To carry
328 out flow cytometry assay, the sequence of copGFP in the pCDH plasmids was deleted via
329 KOD -Plus-Mutagenesis Kit (TOYOBO, SMK-101).

Line 339 The lysis buffer for BCL-2 family proteins was 20 mM Tris-HCl pH 8.0, and 500 mM
NaCl. The lysis buffer for p53 proteins was 20 mM sodium...

Line 346 Untagged Bak was obtained by incubation with PreScission protease at 4°C before Mono
Q

Line 399 The detergent CHAPS alone was reported not to affect the Bax or Bak structure^{61,62}

Response: We appreciate the reviewer's careful reading and valuable feedback. We have made
corresponding modifications to the wording and phrases in the revised manuscript.

Reviewer #2 (Remarks to the Author):

The authors have addressed all my concerns to my full satisfaction. Only a few minor issues remain:

Line 98 - BCL-2 did not influence its binding affinity

Line 149 – I think this should be EC50

Line 152 – did the authors determine Kd or KD with their assays?

Line 305 – artefacts

Line 500 – Protein structure coordinate availability

Figure 3: Comparison of Kd fit:

Figure 3: Mutational analysis

Line 720: Curves were fitted and Kd values of interactions calculated using experimental data with
a Kd model.....

It would be good if the changes are applied to the same issues in the supplementary data (KD vs Kd,
EC50 etc)

Response: We appreciate the reviewer's careful reading and valuable feedback. We have made
corresponding modifications to the wording and phrases in the revised manuscript. The MST was
used to determine KD, and we have revised accordingly. The changes have also been applied to the
same issues in the supplementary data (KD, EC50).